# CPCF: A Cross-Prompt Contrastive Framework for Referring Multimodal Large Language Models

**Lanyun Zhu** [1]  **Deyi Ji** [2]  **Tianrun Chen** [3]  **Haiyang Wu** [2]  **De Wen Soh** [1]  **Jun Liu** [4]

## Abstract

Referring MLLMs extend conventional multimodal large language models by allowing them to receive referring visual prompts and generate responses tailored to the indicated regions. However, these models often suffer from suboptimal performance due to incorrect responses tailored to misleading areas adjacent to or similar to the target region. This work introduces CPCF, a novel framework to address this issue and achieve superior results. CPCF contrasts outputs generated from the indicated visual prompt with those from contrastive prompts sampled from misleading regions, effectively suppressing the influence of erroneous information outside the target region on response generation. To further enhance the effectiveness and efficiency of our framework, several novel designs are proposed, including a prompt extraction network to automatically identify suitable contrastive prompts, a self-training method that leverages unlabeled data to improve training quality, and a distillation approach to reduce the additional computational overhead associated with contrastive decoding. Incorporating these novel designs, CPCF achieves state-of-the-art performance, as demonstrated by extensive experiments across multiple benchmarks. Project page: https://lanyunzhu.site/CPCF/

## 1. Introduction

Recently, multimodal large language models (MLLMs) (Liu et al., 2023; Li et al., 2023b) have established a new paradigm for multimodal learning, achieving remarkable success in tasks such as image captioning and visual question answering. However, conventional MLLMs are typically restricted to only coarse, image-level understanding

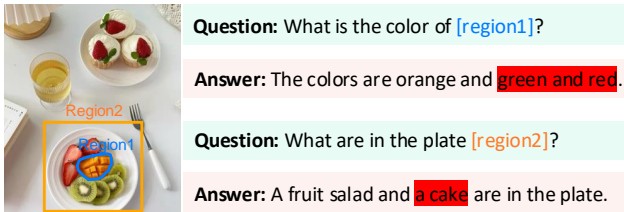

*Figure 1.* An example of errors in existing referring MLLM models caused by misleading regions. The incorrect contents in the responses are highlighted in red.

and struggle with finer-grained visual tasks, such as regional description. To address this limitation, several referring MLLMs (You et al., 2023; Peng et al., 2023) have been proposed. These methods receive various types of visual prompts as input, such as points, bounding boxes, and masks, allowing the MLLM to focus on specific regions and generate responses tailored to the indicated areas.

While these methods have achieved some success, as shown in Figure 1, their performance often remains suboptimal, frequently generating incorrect responses related to misleading areas adjacent to or resembling the target region, rather than being fully tailored to the input visual prompt. For example, in the case shown in Figure 1, the MLLM incorrectly infers the color of region 1 as "green and red", corresponding to the surrounding kiwifruits and strawberries. Additionally, it mistakenly concludes that there is a cake on the plate in region 2, influenced by the presence of another plate in the upper-right corner of the image that contains some cakes. To investigate further, we use image editing tools to modify these misleading image regions (e.g., replacing kiwifruits and strawberries on the plate with blueberries, or replacing the cake on the top-right plate with a steak). We observe that the erroneous content in the MLLM's responses changes accordingly (e.g., from "green and red" to "blue", or from "a cake" to "a steak"), indicating that these errors are due to confusion caused by misleading regions rather than merely being LLM's inherent hallucinations unrelated to the image content. Such errors frequently occur in existing referring MLLM methods, underscoring their limitations in robustness and effectiveness, highlighting the need for developing novel mechanisms to further enhance response reliability.

[1]Singapore University of Technology and Design [2]Tencent [3]Zhejiang University [4]Lancaster University. Correspondence to: Tianrun Chen, Lanyun Zhu <lanyun_zhu@mymail.sutd.edu.sg, tianrun.chen@zju.edu.cn>.

*Proceedings of the $42^{nd}$ International Conference on Machine Learning*, Vancouver, Canada. PMLR 267, 2025. Copyright 2025 by the author(s).

To address the above issues, in this work, we propose a novel and effective referring MLLM framework with enhanced performance, inspired by the success of contrastive decoding (Li et al., 2022; Leng et al., 2023) in reducing hallucinations for LLMs and MLLMs. The core idea of our framework is to contrast outputs generated from different visual prompts—specifically, the original input prompt $r$ and contrastive prompts $\hat{r}$ extracted from misleading regions. This approach amplifies the accurate information associated with $r$ while suppressing erroneous information related to the misleading regions, thereby enhancing response accuracy and reliability. Building such a framework requires addressing several critical technical challenges. The first challenge is how to identify suitable contrastive prompts $\hat{r}$ to achieve high-performance contrastive decoding. A naive and straightforward approach is to randomly and manually sample points from regions adjacent to or similar to $r$ as $\hat{r}$. However, such handcrafted and random methods lack robustness and are difficult to yield optimal prompts, since even small variations in the sampled points (e.g., shifts of 5 pixels) may result in significant changes to the final results. To address this limitation, we propose a novel prompt extraction network that *automatically* finds optimal contrastive prompts based on the input image and instruction, eliminating the uncertainty and instability inherent in manual methods. Furthermore, we introduce a self-training method that leverages unlabeled data and self-generated question-answer pairs for DPO training, effectively optimizing the performance of the prompt extraction network and resulting in improved contrastive decoding outcomes.

Another key challenge is the additional computational cost incurred by the multiple executions of the MLLM during contrastive decoding. To address this issue and improve efficiency, we propose a novel distillation method that transfers the capabilities of the multi-execution contrastive decoding framework to a student model requiring only a single execution, thereby significantly reducing computational cost. To enhance the effectiveness of this distillation process, a diffusion-based inpainting loss function is further introduced, ensuring regional contrastive information to be effectively extracted and utilized by the student model. Incorporating these designs, we develop a novel and effective referring MLLM, CPCF, with extensive experiments across multiple benchmarks and tasks demonstrating its outstanding performance and significant advantages.

In conclusion, the main contributions of our work are as follows: (1) We introduce CPCF, an effective referring MLLM framework that contrasts input prompts with contrastive prompts from misleading regions, enabling highly accurate region understanding and high-quality text response generation. (2) We propose several novel designs to enhance model performance and efficiency, including an automatic prompt extraction mechanism to identify optimal contrastive prompts, a self-training method to improve network optimization, and the first distillation framework tailored for contrastive decoding techniques to reduce computational costs. (3) Extensive experiments on multiple benchmarks demonstrate that our CPCF achieves SOTA performance.

## 2. Related Work

**Referring Multimodal Large Language Models.** Recent advancements in multimodal large language models (MLLMs) (Li et al., 2023b; Liu et al., 2023; Zhu et al., 2023; Li et al., 2023a; Zhu et al., 2024a; 2025b;a; Ji et al., 2024; Chen et al., 2024a) have expanded the capabilities of conventional LLMs (Touvron et al., 2023; Bai et al., 2023) to the visual domain, achieving remarkable performance on tasks such as image captioning and visual question answering. Referring MLLMs (You et al., 2023; Zhang et al., 2024; Wu et al., 2024; Zhang et al., 2023; Peng et al., 2023; Chen et al., 2023; Xuan et al., 2024; Yue et al., 2024; Yuan et al., 2024; Rasheed et al., 2024) further enable models to receive prompts in the form of points, boxes, or masks as input and generate responses for the indicated regions, allowing users to interact with models in a more fine-grained manner. However, existing referring MLLM methods often produce incorrect answers by mistakenly focusing on confusing regions adjacent to or similar to the indicated referring regions. To address this limitation, this work introduces a novel, task-specific framework that mitigates these errors and significantly improves performance.

**Contrastive Decoding.** To address the hallucination problem in LLMs, contrastive decoding techniques have been proposed to generate more accurate responses by comparing outputs from different models (Li et al., 2022) or different inputs (Kim et al., 2024). Recently, contrastive decoding has been extended to MLLMs (Leng et al., 2023; Zhu et al., 2024b; Wang et al., 2024; Chen et al., 2024b) to improve their reliability in multimodal scenarios. However, most existing contrastive decoding methods are not specifically designed for referring MLLMs. The most closely related work to ours is CRG (Wan et al., 2025), which also employs contrastive decoding for referring tasks. However, CRG relies on perturbing certain regions of the image to generate contrastive inputs, a process that can introduce noise and errors. Moreover, CRG incurs significant additional computational costs, as it requires running the model multiple times for each input. In contrast, our method utilizes different prompts as contrastive counterparts, avoiding the semantic disruption caused by CRG's perturbation-based approach. Additionally, a novel mechanism is proposed to automatically generate contrastive prompts, addressing the instability associated with manual strategies. Furthermore, we introduce an innovative distillation method that substantially reduces the computational overhead typically associated with most existing contrastive decoding meth-

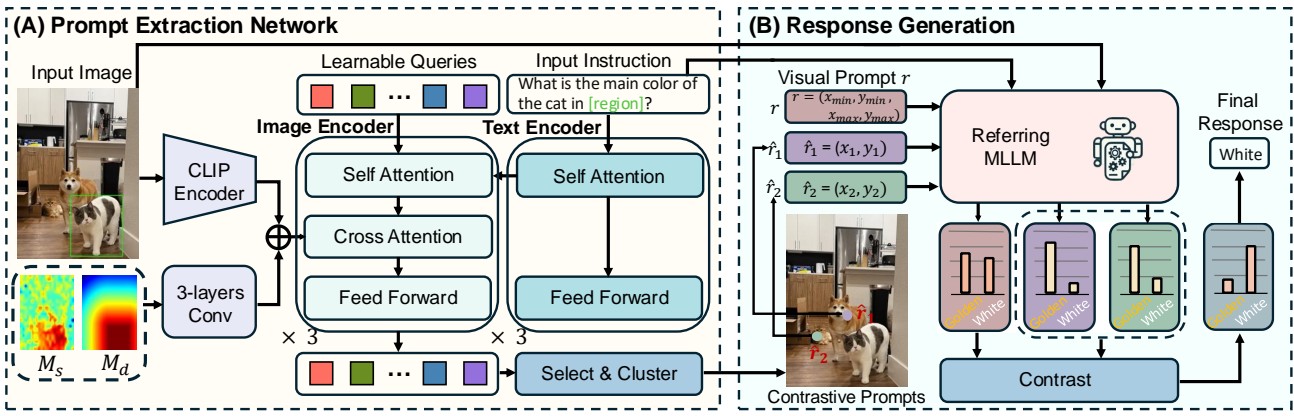

Figure 2. Overall pipelines of (A) prompt extraction network and (B) response generation process.

ods, including CRG. Incorporating these novel designs, our framework is significantly different from CRG. Results in Table 3 highlight the performance advantages of our method over existing contrastive decoding techniques.

## 3. Method

### 3.1. Preliminaries and Overview

Referring MLLMs extend standard MLLMs by incorporating an additional prompt, $r$, as input to help the model focus on specific regions of interest[1]. The prompt $r$ can take various forms, such as bounding boxes, masks, or points. The prediction process of a referring MLLM is formally defined as: $p(.|y_i) = \text{Softmax}[\text{MLLM}(I, q, x, y_{<i})]$, where $I$ represents the input image, $x$ is the instruction, $y_{<i}$ is the first $i-1$ tokens in the response $y$, and $p$ denotes the prediction likelihood. Building on the basic structure of Ferret (You et al., 2023), our method incorporates a contrastive decoding framework to mitigate a common issue in existing methods: misfocusing on confusing regions. Formally, the prediction process of the proposed approach can be written as:

$$p(.|y_{<i}) = \text{Softmax}[\frac{1}{N_{\hat{r}}} \sum_{n=1}^{N_{\hat{r}}} ((1+\alpha)\text{MLLM}(I, x, r, y_{<i}) \\ -\alpha\text{MLLM}(I, x, \hat{r}_n, y_{<i}))], \tag{1}$$

where $\hat{r}_n$ denotes the $n$-th contrastive referring prompt (in the form of points) and $\alpha$ is a hyperparameter. To improve performance and enhance efficiency, we propose a three-step training process for our framework: (1) First, the MLLM is pretrained using the same method as Ferret. (2) Next, the pretrained MLLM is frozen, and a prompt extraction network (**Sec.3.2**) is introduced to generate the contrastive prompts $\{\hat{r}_n\}_{n=1}^{N_{\hat{r}}}$ in Eq.1. This network is trained using

---

[1]In the main paper, we illustrate our method for the inputs containing only one visual prompt $r$. Our method can be easily extended to multi-$r$ scenarios, as introduced in Appendix Sec.A.1.

the approach described in **Sec.3.3**. The combined model, consisting of the MLLM and the prompt extraction network, is denoted as $\phi_m$. (3) Finally, to reduce the additional computational overhead introduced by contrastive decoding, the trained $\phi_m$ is distilled into $\phi_s$ (**Sec.3.4**), which executes the MLLM only once. The following sections provide a detailed explanation of the key techniques in our framework.

### 3.2. Generation of Contrastive Referring Prompts

**Prompt Extraction Network.** We propose a prompt extraction network to automatically generate the contrastive referring prompts, instead of relying on manual strategies, which are often suboptimal and lack robustness (as discussed in the Introduction). As illustrated in Figure 2 (A), the network comprises an image encoder and a text encoder, each with 3 stages. The input to the image encoder consists of 64 learnable query embeddings $e = \{e_i\}_{e=1}^{64}$, while the input to the text encoder is the instruction $x$. In each stage, the hidden states of $e$ first interact with both themselves and the text tokens from the text encoder via self-attention. They then interact with image tokens through cross-attention, where the image tokens are enhanced by prior information (detailed in the next section). The hidden states of each $e_i$ from the final stage are passed through a linear layer to produce 2D coordinates $(h_i, w_i)$ and a confidence score $c_i$. Notably, the text features of the instruction are integrated into the prompt generation process. This design ensures that the generation of the contrastive prompt is instruction-dependent, as different instructions may cause the model to mistakenly focus on different confusing regions. We utilize the first 3 stages of the Q-Former from BLIP2 (Li et al., 2023b) to initialize the network, leveraging its pre-learned, effective image-text feature interaction capabilities.

**Enhancement with Prior Information.** As discussed in the Introduction, referring MLLMs often mistakenly focus on regions adjacent to or similar to $r$' corresponding region **R**. To assist the prompt extraction network in generating the

more appropriate $\hat{r}$, we incorporate these empirical observations as prior information to enhance the generation process. We begin by creating two masks: a semantic similarity map $M_s \in \mathbb{R}^{H \times W}$ and a relative distance map $M_d \in \mathbb{R}^{H \times W}$, where $H$ and $W$ are the height and width of the input image $I$. Each pixel $M_s^{i,j}$ in $M_s$ is computed as the L2 distance between the CLIP features of $(i, j)$ and the average features of $\mathbf{R}$. Each $M_d^{i,j}$ in $M_d$ represents the coordinate Euclidean distance from $(i, j)$ to its nearest pixel within $\mathbf{R}$. We normalize $M_s$ and $M_d$ to the range [0, 1], with the pixels inside $\mathbf{R}$ set to -1 as a flag indicating region $\mathbf{R}$. $M_s$ and $M_d$ are then processed through a 3-layer CNN, producing outputs $f_s$ and $f_d$, which are added to the image tokens. The enhanced tokens are finally utilized in the image encoder of the prompt extraction network, where they interact with the query embeddings $e$ (see the previous section).

**Contrastive Prompt Selection.** We select appropriate prompts from the 64 outputs $\{h_i, w_i, c_i\}_{i=1}^{64}$ generated by the prompt extraction network. To enhance exploratory behavior and improve robustness, we add a random standard Gaussian noise $\mathcal{N}$ to $c_i$ and compute $s_i = \text{Sigmoid}(c_i + \mathcal{N})$. Points $(h_i, w_i)$ with $s_i > 0.5$ are considered valid contrastive prompts. We observe that many of these prompts are located in adjacent positions. To reduce redundancy, we apply the Mean Shift algorithm to cluster the positions of all valid prompts. The resulting cluster centers are then used as the contrastive prompts $\{\hat{r}_n\}_{n=1}^{N_{\hat{r}}}$ in Eq.1.

### 3.3. Optimization with Self-Training

**Motivation.** The next key challenge is how to train the prompt extraction network effectively. We first tried two straightforward approaches: (1) we initialize the MLLM parameters using LLaVA and jointly train both the MLLM and the prompt extraction network on the referring question-answer dataset $\mathcal{D}$. However, this approach results in unstable training and poor performance. (2) We first train the MLLM on $\mathcal{D}$ following Ferret's method. Afterward, the MLLM is frozen, and the prompt extraction network is added and trained on the same dataset $\mathcal{D}$. While this leads to some improvement, the results remain unsatisfactory. A potential reason for the limited success is that the MLLM, having already trained on $\mathcal{D}$ and fit well to it, exhibits few hallucinations and errors on the training set. As a result, there is very minimal room for further enhancement through contrastive decoding on $\mathcal{D}$, making it difficult to train the prompt extraction network effectively. To address this, we introduce an additional dataset $\tilde{\mathcal{D}}$ beyond $\mathcal{D}$ — sampled from the Open Images dataset (Kuznetsova et al., 2020) — to train the prompt extraction network, which is not used in MLLM's training and contains rich bounding box labels. To mitigate the issue that $\tilde{\mathcal{D}}$ lacks question-answer annotations, we propose a self-training mechanism, in which the MLLM, after trained on $\mathcal{D}$, is prompted to generate questions and

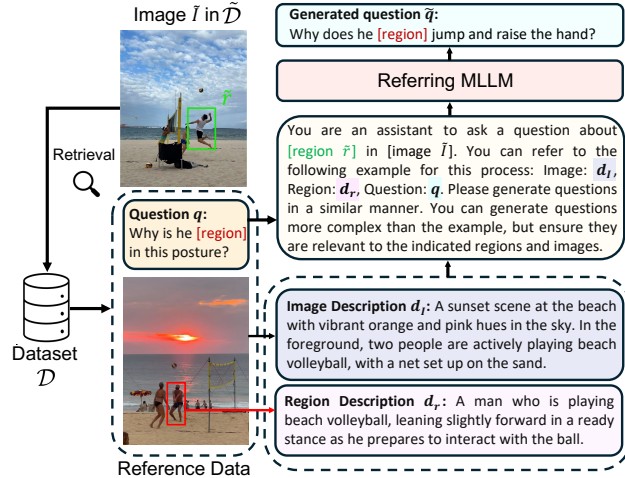

*Figure 3.* Illustration of the RAG-based question generation process in the proposed self-training method.

answers for each image in $\tilde{\mathcal{D}}$ to train the prompt extraction network, enabling instruction tuning without the costly and labor-intensive manual or GPT-based data annotation.

**Question Generation with RAG.** We denote each data $D^i$ in the Ferret's training set $\mathcal{D}$[2] as $D^i = \{I^i, r^i, q^i, a^i\}$, representing the image, referring prompt, question, and answer, respectively. The CLIP feature of $I^i$ is computed as the global representation $f_g^i$, and the average CLIP features of region $r^i$ serve as a local representation $f_l^i$. For each image $\tilde{I}$ in the new set $\tilde{\mathcal{D}}$, we randomly select a labeled bounding box and randomly transform it into a point or mask, creating the prompt $\tilde{r}$. The corresponding global and local representations $\tilde{f}_g$ and $\tilde{f}_l$ for $\tilde{I}$ and $\tilde{r}$ are then computed. We calculate the cosine similarity $s_g^i$ between $\tilde{f}_g$ and each $f_g^i$ in $\mathcal{D}$, as well as $s_l^i$ between $\tilde{f}_l$ and each $f_l^i$. The data $D^i$ with the highest $s_g^i + s_l^i$ among $\mathcal{D}$ is selected, and two text descriptions, $d_I$ and $d_r$, are then generated by the MLLM for the full image $I^i$ and the region $r^i$ in the selected $D^i$, respectively (see Appendix Sec.A.2 for the prompt). These descriptions, along with the corresponding question $q$, form an in-context example that prompts the MLLM to generate a question $\tilde{q}$ for the region $\tilde{r}$ on $\tilde{I}$ in a similar fashion. Please refer to Figure 3 for the detailed prompt used in this process. Notably, we employ a retrieval-augmented generation (RAG) approach with in-context learning for this question generation method. This is because directly prompting the MLLM to generate questions often results in errors and hallucinations, producing questions unrelated to $\tilde{I}$ and $\tilde{r}$. In contrast, providing the MLLM with a similar example from $\mathcal{D}$ for reference can significantly improve reliability.

---

[2]The data from the VCR dataset within $\mathcal{D}$ is used for RAG because it contains diverse and complex questions.

Results in Sec.4.3 and Appendix Sec.C.1 demonstrate the effectiveness of the proposed approach.

**Answer Generation and Preference Optimization.** With the image-prompt pair $\{\tilde{I}, \tilde{r}\}$ and the generated question $\tilde{q}$, the MLLM is further prompted to generate two answers, $\tilde{a}$ and $\tilde{a}^{cot}$. $\tilde{a}$ is directly obtained by $\mathrm{MLLM}(\tilde{I}, \tilde{r}, \tilde{q})$, while $\tilde{a}^{cot}$ is produced by incorporating a chain-of-thought (CoT) prompt $p_c$, i.e., $\mathrm{MLLM}(\tilde{I}, \tilde{r}, \tilde{q}, p_c)$, encouraging the MLLM to generate results in a more detailed, step-by-step manner (see Appendix Sec.A.3 for details of $p_c$). Results presented in the Appendix reveal that $\tilde{a}^{cot}$ is more accurate than $\tilde{a}$ in most cases. Based on this observation, we employ the following direct preference optimization (DPO) loss (Rafailov et al., 2024) to train the prompt extraction network:

$$\mathcal{L}_{dpo} = -\mathbb{E}_{\tilde{\mathcal{D}}}[\log \sigma(\beta \log \frac{\pi_\theta(\tilde{a}^{cot}|\tilde{I}, \tilde{r}, \tilde{q})}{\pi_{\mathrm{ref}}(\tilde{a}^{cot}|\tilde{I}, \tilde{r}, \tilde{q})}$$
$$-\beta \log \frac{\pi_\theta(\tilde{a}|\tilde{I}, \tilde{r}, \tilde{q})}{\pi_{\mathrm{ref}}(\tilde{a}|\tilde{I}, \tilde{r}, \tilde{q})})], \quad (2)$$

where $\beta$ is a hyperparameter set to 0.5, $\pi_\theta$ is the policy MLLM that is continuously updated during training, and $\pi_{\mathrm{ref}}$ is a reference MLLM updated via a Momentum strategy in each iteration, i.e., $\pi_{\mathrm{ref}} = 0.999\pi_{\mathrm{ref}} + 0.001\pi_\theta$. $\log \pi(\tilde{a}|\tilde{I}, \tilde{r}, \tilde{q}) = \frac{1}{|\tilde{a}|} \sum_{i=1}^{|\tilde{a}|} \log p(\tilde{a}_i|\tilde{I}, \tilde{r}, \tilde{q}, \tilde{a}_{<i})$ is the average likelihood of all tokens in $\tilde{a}$ computed by Eq.1.

### 3.4. Distillation for Computation Reduction

**Motivation.** The aforementioned model structures and training methods enable effective contrastive decoding with enhanced performance. However, this contrastive decoding method introduces significant computational overhead in inference, as the MLLM needs to be executed multiple times for different contrastive referring prompts to generate the response for each input $(I, r, x)$ (see Eq.1). To reduce computational complexity, inspired by the success of knowledge distillation in large models (Wan et al., 2023), we propose distilling the results of the original *multi-execution model* $\phi_m$, trained on $\tilde{\mathcal{D}}$, into a *single-execution model* $\phi_s$, which requires only one pass of the MLLM for each input $(I, r, x)$, thus reducing computational cost.

**Adapter Module for Single-Execution Model.** As shown in Figure 4, the single-execution model $\phi_s$ retains the core structure of $\phi_m$, incorporating a prompt extraction network and an MLLM. To facilitate finetuning during the distillation process, 4 adapter modules are inserted at evenly distributed intervals throughout the LLM[3] in $\phi_s$. As illustrated in Figure 4, each adapter module consists of $N_a$ tokens $\mathbf{A} = \{A_i\}_{i=1}^{N_a}$, which are concatenated with the hidden states output by the preceding LLM layer and processed

---

[3] At the 8th, 16th, 24th, and 32nd layers of the 7B model, and the 10th, 20th, 30th, and 40th layers of the 13B model

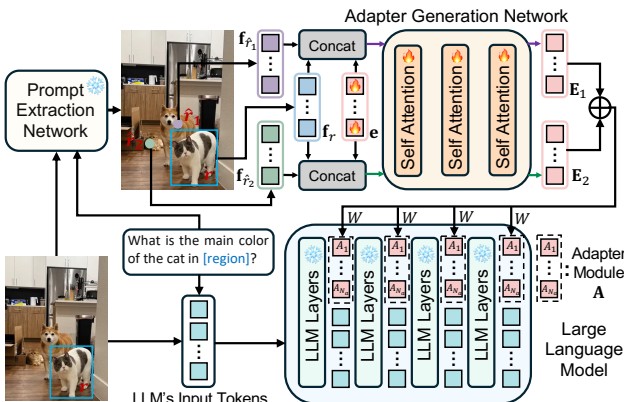

*Figure 4.* Model structure of the single-execution model $\phi_s$ that is distilled from the multi-execution model.

by the attentions in the current layer. To generate $\mathbf{A}$, we first extract regional features $\mathbf{f}_r$ (with 32 tokens) using Ferret's method (see (You et al., 2023) Sec 3.2) for the input prompt $r$'s region $\mathbf{R}$. The same method is employed to generate $\mathbf{f}_{\hat{r}_n} \in \{\mathbf{f}_{\hat{r}_n}\}_{n=1}^{N_{\hat{r}}}$ for each contrastive prompt $\hat{r}_n \in \{\hat{r}_n\}_{n=1}^{N_{\hat{r}}}$. The adapter tokens $\mathbf{A}$ are then computed as:

$$[; ; \mathbf{E}_n] = \mathrm{AGN}\left(\mathrm{Concat}\left[\mathbf{f}_r; \mathbf{f}_{\hat{r}_n}; \mathbf{e}\right]\right), \mathbf{A} = W\left(\sum_{n=1}^{N_{\hat{r}}} \mathbf{E}_n\right), \quad (3)$$

where AGN is an adapter generation network composed of 3 sequential self-attention layers, $\mathbf{e} = \{e_i\}_{i=1}^{N_a}$ is a set of learnable embeddings and $\mathbf{A}$ refers to their hidden states at AGN's output layer, $W$ denotes a projection layer. To reduce computational and parameter overhead, the adapter modules at different LLM layers share the same AGN but use independent projection layers $W$.

**Distillation Loss.** We use the trained multi-execution model $\phi_m$ to distill the adapter-based single-execution model $\phi_s$ on $\tilde{\mathcal{D}}$. During this process, the parameters of the prompt extraction network and MLLM in $\phi_s$ are copied from $\phi_m$ and frozen, while only the adapter generation network AGN and projection layers $W$ are updated. The distillation loss function $\mathcal{L}_{dis}$ is calculated as:

$$\mathcal{L}_{dis} = \frac{1}{|\tilde{a}^{cot}|} \sum_{i=1}^{|\tilde{a}^{cot}|} \mathrm{KL}\left(p_{\phi_m}(\cdot|\tilde{a}^{cot}_{<i}) \| p_{\phi_s}(\cdot|\tilde{a}^{cot}_{<i})\right), \quad (4)$$

where KL refers to Kullback-Leibler divergence. $\tilde{a}^{cot}$ denotes each generated answer in dataset $\tilde{\mathcal{D}}$ (see Sec 3.3), $p_{\phi_m}(\cdot|\tilde{a}^{cot}_{<i})$ refers to the prediction probability distribution in the $i$-th time step of $\phi_m$ obtained by Eq.1. $p_{\phi_s}(\cdot|\tilde{a}^{cot}_{<i})$ is directly generated in a single step, i.e., $\mathrm{Softmax}[\mathrm{MLLM}(I, q, r, \tilde{a}^{cot}_{<i})]$.

**Inpainting-Based AGN Optimization.** In addition to $\mathcal{L}_{dis}$, we also propose an inpainting loss to optimize the adapter

generation network AGN. The motivation stems from the fact that contrastive decoding in $\phi_m$ enhances performance by comparing the regions associated with the input prompt $r$ against those corresponding to different contrastive prompts $\{\hat{r}_n\}_{n=1}^{N_{\hat{r}}}$. Therefore, to effectively distill $\phi_m$ into $\phi_s$, the adapter modules integrated into the MLLM of $\phi_s$ are optimized to also capture and encode these regional differences. For each training image $I$, let $\mathbf{R}$ represent the region corresponding to the input referring prompt $r$. We construct a region $\hat{\mathbf{R}}_n$ with the same shape as $\mathbf{R}$, centered at each contrastive prompt $\hat{r}_n$, and replace $\hat{\mathbf{R}}_n$ in $I$ with $\mathbf{R}$ to generate a perturbed image $\hat{I}_n$. $\hat{I}_n$ is inpainted back to $I$ using a frozen stable diffusion model SD. The inpainting-based loss function $\mathcal{L}_{inp}$ is then calculated as:

$$p_{pos} = d_{\hat{\mathbf{R}}_n}, \ p_{neg}^n = L(\mathbf{E}_n), \ \bar{I}_n = \text{SD}\left(\hat{I}_n, \hat{\mathbf{R}}_n, p_{pos}, p_{neg}^n\right),$$

$$\mathcal{L}_{inp} = \mathbb{E}_n\left(||\bar{I}_n - I||^2 - ||\bar{I}_n - \hat{I}_n||^2\right),$$

(5)

where $d_{\hat{\mathbf{R}}_n}$ refer to a textual description for region $\mathbf{R}_n$ generated by the referring MLLM. $L$ denotes a linear layer, and its input $\mathbf{E}_n$ is generated by AGN through Eq.3. $p_{pos}$ and $p_{neg}$ respectively refers to the positive and negative prompt used in stable diffusion. The idea is straightforward: $\mathcal{L}_{inp}$ can be minimized if $\mathbf{E}_n$ contains differential information that describes what is present in $\mathbf{R}$ but absent in $\hat{\mathbf{R}}_n$. This information can serve as a negative prompt to prevent the inpainting result of $\mathbf{R} \rightarrow \hat{\mathbf{R}}_n$ (during $\hat{I}_n \rightarrow I$) from preserving the characteristics of $\mathbf{R}$. By leveraging $\mathcal{L}_{inp}$, AGN can be optimized to extract more comprehensive and accurate regional difference information, facilitating the distillation process from $\phi_m$ to $\phi_s$. Finally, the overall distillation loss is computed as the sum of $\mathcal{L}_{dis}$ (Eq.4) and $\mathcal{L}_{inp}$ (Eq.5).

## 4. Experiments

### 4.1. Implementation Details

We follow the basic model structures of Ferret, using CLIP-ViT-L/14@336p as the image encoder and Vicuna as the LLM. As described in Sec.3.1, the entire training process consists of three steps. In the first step, initial training is conducted in the same manner as Ferret. In the second step, from the Open Images dataset (Kuznetsova et al., 2020), we sample 50K images that were not used in the first step and are not included in the test set, synthesize corresponding questions and answers to generate dataset $\tilde{\mathcal{D}}$, and train the prompt extraction network for 50K steps (details in Sec.3.3). This step takes about 30/50 hours for the 7B/13B models on 8 A100 GPUs. In the third step, the model is distilled into a single-execution model $\phi_s$ by training for 50K steps on $\tilde{\mathcal{D}}$ (details in Sec.3.4). This step takes about 25/40 hours for the 7B/13B models. The number $N_a$ of tokens in each adapter module of model $\phi_s$ is set to 32. $\alpha$ in Eq.1 is 0.5.

### 4.2. Comparison with Other Methods

**Evaluated Tasks and Benchmarks.** Following prior work (You et al., 2023; Wu et al., 2024), we evaluate our approach against other methods across four tasks and benchmarks: *Referring Object Classification (ROC)*: This task requires the model to identify the object within the referring region. We use the benchmark provided by (Wu et al., 2024), which generates 1748 questions from the LVIS validation set (Gupta et al., 2019) for evaluation. *(2) Referring Text Classification (RTC)*: In this task, the model selects the text candidate corresponding to the referring object from several options. The evaluated benchmark is from (Wu et al., 2024), generated from the COCO-Text dataset (Veit et al., 2016). *(3) Referring Description (RD)*: This task is evaluated on *RefCOCOg* (Kazemzadeh et al., 2014), requiring the model to generate a textual description of the referring bounding box region. To accelerate testing, 1000 samples are randomly selected from this benchmark for evaluation. *(4) Referring Reasoning and Referring Captioning*: These tasks are evaluated on *Ferret-Bench*, as proposed by (You et al., 2023), which includes more challenging description and reasoning problems. Please refer to Appendix Sec.B for more details, including evaluation metrics and input instructions used during testing.

**Comparison Results.** Results for the Referring Object Classification (ROC), Referring Text Classification (RTC) and Referring Description (RD) tasks are presented in Table 1, and comparison results for the Referring Reasoning and Referring Captioning tasks are shown in Table 2. Compared to other referring MLLM approaches such as Ferret (You et al., 2023), our method consistently achieves the best performance across all benchmarks with significant advantages, demonstrating the excellent effectiveness of our approach and its strong generalization capability across diverse tasks.

**Evaluation of Errors Related to Misleading Regions.** In *Appendix Sec.C.2*, we provide a more detailed comparison with other methods, specifically focusing on errors caused by the model incorrectly misled by regions adjacent to or similar to the input visual prompt.

**Comparison with Other Contrastive Decoding Methods.** Based on the Ferret model, we further compare our method with other contrastive decoding approaches on the referring object classification task. As shown in Table 3, benefiting from our task-tailored designs and the automatically learned contrastive prompts, our method achieves the best performance, demonstrating its significant advantage over existing contrastive decoding methods in addressing referring tasks.

### 4.3. Ablation Study

We conduct ablation study on the referring object classification (ROC) task. Due to paper length limitation, more

*Table 1.* Comparison results on the **Referring Object Classification (ROC)** task, **Referring Text Classification (RTC)** task, and **Referring Description (RD)** task.

| Models | ROC | | | | RTC | | RD | | | |
|---|---|---|---|---|---|---|---|---|---|---|
| | Box | Mask | Scribble | Point | Box | Mask | B@4 | M | C | S |
| Kosmos2-1.6B (Peng et al., 2023) | 55.17 | - | - | - | 16.55 | - | 38.68 | 23.90 | 66.99 | 19.76 |
| GPT4RoI-7B (Zhang et al., 2023) | 58.59 | - | - | - | 54.23 | - | 36.12 | 24.83 | 69.17 | 20.05 |
| Shikra-7B (Chen et al., 2023) | 64.60 | - | - | 56.27 | 50.07 | - | 36.55 | 26.43 | 73.80 | 20.09 |
| CogVLM-17B (Wang et al., 2023) | 68.44 | - | - | - | 52.29 | - | 39.05 | 27.75 | 75.86 | 20.51 |
| GLaMM-7B (Rasheed et al., 2024) | 69.93 | - | - | - | 53.92 | - | 40.12 | 28.08 | 75.80 | 21.75 |
| Osprey-7B (Yuan et al., 2024) | 72.15 | 74.19 | - | - | 55.98 | 60.45 | 40.50 | 27.33 | 79.18 | 21.47 |
| Ferret-7B (You et al., 2023) | 71.71 | 72.39 | 71.58 | 68.54 | 55.47 | 56.34 | 41.30 | 27.18 | 78.36 | 21.62 |
| Ferret-13B (You et al., 2023) | 72.83 | 73.75 | 72.33 | 69.70 | 56.24 | 58.90 | 41.78 | 28.22 | 80.24 | 21.91 |
| **CPCF-7B (Ours)** | **78.37** | **79.55** | **77.97** | **77.14** | **62.69** | **63.38** | **44.23** | **29.79** | **82.41** | **22.67** |
| **CPCF-13B (Ours)** | **79.09** | **80.72** | **78.85** | **78.19** | **63.90** | **64.28** | **45.11** | **30.55** | **84.52** | **23.07** |

*Table 2.* Comparison results on the **Referring Reasoning** task and **Referring Captioning** tasks within the Ferret-Bench benchmark.

| Models | Referring Captioning | Referring Reasoning |
|---|---|---|
| Kosmos2-1.6B (Peng et al., 2023) | 51.8 | 33.7 |
| Shikra-7B (Chen et al., 2023) | 46.0 | 41.6 |
| CogVLM-17B (Wang et al., 2023) | 67.1 | 67.6 |
| Osprey-7B (Yuan et al., 2024) | 72.2 | 67.8 |
| Ferret-7B (You et al., 2023) | 68.7 | 67.3 |
| Ferret-13B (You et al., 2023) | 70.6 | 68.7 |
| Ferret-v2-7B (Zhang et al., 2024) | 79.9 | 81.7 |
| Ferret-v2-13B (Zhang et al., 2024) | 79.6 | 79.4 |
| **CPCF-7B (Ours)** | **83.5** | **82.9** |
| **CPCF-13B (Ours)** | **84.0** | **83.2** |

*Table 3.* Comparison results with other contrastive decoding methods on the Referring Object Classification (ROC) task.

| Models | Box | Mask | Scribble | Point |
|---|---|---|---|---|
| Ferret-7B | 71.71 | 72.39 | 71.58 | 69.54 |
| Ferret-7B + VCD (Leng et al., 2023) | 74.90 | 75.08 | 73.04 | 72.40 |
| Ferret-7B + ICD (Wang et al., 2024) | 74.55 | 75.73 | 73.58 | 72.05 |
| Ferret-7B + CRG (Wan et al., 2025) | 75.40 | 76.20 | 75.24 | 73.39 |
| **CPCF-7B (Ours)** | **78.37** | **79.55** | **77.97** | **77.14** |

*Table 4.* Ablation study of key components in our framework.

| Methods | Box | Mask | Inference Time (Method / Ours) |
|---|---|---|---|
| Ours | 78.37 | 79.55 | 1.00 |
| Ours w/o prompt extraction network | 74.33 | 75.08 | 0.98 |
| Ours w/o self-training | 73.19 | 74.90 | 1.00 |
| Ours w/o distillation | 79.08 | 80.15 | 3.79 |

*Table 5.* Ablation study of prompt extraction network.

| Methods | Box | Mask | Scribble | Point |
|---|---|---|---|---|
| Ours | 78.37 | 79.55 | 77.97 | 77.14 |
| Ours w/o initialization from BLIP2 | 76.93 | 77.88 | 76.44 | 75.90 |
| Ours w/o $M_s$ | 76.50 | 77.94 | 76.13 | 75.83 |
| Ours w/o $M_d$ | 75.77 | 78.02 | 76.09 | 75.41 |
| Ours w/o random noise for selection | 77.51 | 78.49 | 77.04 | 76.06 |

results are presented in ***Appendix***.

**Effectiveness of Key Components.** We conduct experiments to verify the key components of our framework, with results presented in Table 4. As shown in the table, the model's performance significantly decreases under the following conditions: (1) Excluding the prompt extraction network (Sec.3.2) and using a prior-based manual method to generate the contrastive prompts in Eq.1. (2) Removing the proposed self-training method (Sec.3.3) and directly training the prompt extraction network on the original dataset $\mathcal{D}$. We also evaluate (3) Removing the distillation method (Sec.3.4) and directly using the multi-execution approach described in Eq.1 for testing. Although the model's per-

formance becomes slightly better, the computational cost greatly increases by more than threefold. These results demonstrate the effectiveness of our methods in improving performance and reducing computational overhead.

**Ablation Study of Prompt Extraction Network.** We propose several designs to enhance the effectiveness of the prompt extraction network (Sec.3.2) in generating contrastive prompts, including: (1) Initializing weight parameters using BLIP2's Q-former. (2) Incorporating the semantic similarity map $M_s$ and relative distance map $M_d$ as prior information. (3) Adding random standard Gaussian noise $\mathcal{N}$ for more robust prompt selection. As shown in Table 5, removing any of these design components leads to a significant decline in performance, demonstrating the rationale and effectiveness of our proposed designs in this network.

**Ablation Study of Self-Training.** We further evaluate the designs in the proposed self-training method (Sec.3.3) through the following experiments: (1) Generating questions for the data in $\tilde{\mathcal{D}}$ directly, without using the RAG method. (2) Removing either the global similarity $s_g^i$ or the local similarity $s_l^i$ during retrieval. (3) Replacing the DPO

Table 6. Ablation study of self training.

| Methods | Box | Mask | Scribble | Point |
|---|---|---|---|---|
| Ours | 78.37 | 79.55 | 77.97 | 77.14 |
| Ours w/o RAG for question generation | 75.98 | 76.52 | 75.41 | 75.03 |
| Ours w/o $s_g^i$ for retrieval | 76.99 | 78.22 | 76.68 | 76.51 |
| Ours w/o $s_l^i$ for retrieval | 77.71 | 78.80 | 77.55 | 76.70 |
| Ours w/o DPO w/ CE loss | 75.47 | 76.08 | 75.22 | 75.01 |

Table 7. Effectiveness of different methods to leverage $\tilde{\mathcal{D}}$.

| Methods | Box | Mask | Scribble | Point |
|---|---|---|---|---|
| Baseline Ferret (w/o using $\tilde{\mathcal{D}}$) | 71.71 | 72.39 | 71.58 | 68.54 |
| Use $\tilde{\mathcal{D}}$ through our method (Sec.3.3) | 78.37 | 79.55 | 77.97 | 77.14 |
| Use $\tilde{\mathcal{D}}$ to finetune Ferret | 74.22 | 74.37 | 73.97 | 70.70 |
| Ferret + CRG (w/o fine-tuning) | 75.40 | 76.20 | 75.33 | 74.89 |
| Ferret + CRG (w/ fine-tuning on $\tilde{\mathcal{D}}$) | 75.95 | 77.06 | 75.90 | 75.37 |

Table 8. Ablation study of distillation.

| Methods | Box | Mask | Scribble | Point |
|---|---|---|---|---|
| Ours | 78.37 | 79.55 | 77.97 | 77.14 |
| Ours w/o AGN | 74.71 | 75.83 | 74.08 | 73.73 |
| Ours w/o $\mathcal{L}_{dis}$ (Eq.4) | 72.56 | 74.10 | 71.39 | 71.45 |
| Ours w/o $\mathcal{L}_{inp}$ (Eq.5) | 75.95 | 76.80 | 75.52 | 74.49 |
| Ours w/o distillation w/ direct training | 74.26 | 74.90 | 73.98 | 72.10 |

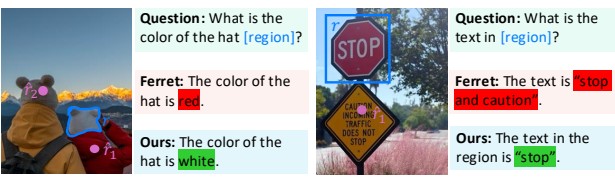

Figure 5. Visualization results of the contrastive prompts generated by the prompt extraction network and the quantitative comparison between our method and Ferret. The contrastive prompts are represented by purple dots. The incorrect contents generated by Ferret and the corresponding correct contents generated by our method are highlighted in red and green, respectively.

### 4.4. Discussion of Computation and Parameter

Compared to the baseline Ferret, our method introduces additional computational and parameter costs due to the prompt extraction network and adapter generation network. However, both modules are highly lightweight, resulting in only a 2.1% increase in parameter count and a 4.3% increase in average inference time compared to Ferret, while achieving significantly improved performance (see Table 1). Moreover, compared to directly using the multi-execution model without distillation, our model requires only 26.4% of its average inference time as the MLLM needs to be executed only once, while maintaining highly comparable performance (see Table 4 "ours w/o distillation"). This highlights the high efficiency and effectiveness of our method.

### 4.5. Visualization Results

In Figure 5, we present two types of visualization results: (1) the contrastive prompts generated by the prompt extraction network; and (2) the quantitative comparison of responses generated by our method and Ferret. In these examples, the generated contrastive prompts are primarily located in misleading regions adjacent to the target region (e.g., clothing next to a hat) or similar to it (e.g., another person's hat). Ferret frequently generates incorrect answers related to these misleading regions rather than completely tailored to the target region (e.g., in the left example, misjudging the hat's color as the color of the adjacent clothing). In contrast, our framework, enhanced by the proposed cross-prompt contrastive methods, significantly improves performance and reduces errors. These results highlight the advantages of our approach over Ferret. Due to space limitations, more

in Eq.2 with a conventional cross-entropy loss, using $\hat{a}^{cot}$ as a pseudo-label. As shown in Table 6, each of these modifications to the original method can result in a performance drop, demonstrating the effectiveness of our designs.

**Different Methods to Leverage $\tilde{\mathcal{D}}$.** In our self-training method, we generate question-answers from $\tilde{\mathcal{D}}$ to optimize the prompt extraction network, thereby enhancing the effectiveness of contrastive decoding. We also test directly finetuning the Ferret on $\tilde{\mathcal{D}}$ without utilizing contrastive decoding and the prompt extraction network. As shown in Table 7, while this approach achieves performance slightly better than the baseline Ferret, it still remains significantly lower than our method for leveraging $\tilde{\mathcal{D}}$. This demonstrates that the improvement of our method is not simply due to the additional training data provided by $\tilde{\mathcal{D}}$, but is primarily attributed to our carefully designed approach that leverages $\tilde{\mathcal{D}}$ to optimize the prompt extraction network and enhance the effectiveness of contrastive decoding. We also fine-tuning Ferret-7B on $\tilde{\mathcal{D}}$ using the contrastive decoding method CRG. As shown in the table below, this method achieves an accuracy of 75.95% on the ROC-Box scenario, outperforming baseline Ferret (71.71%) and naive fine-tuning (74.22%), but still falling significantly short of our CPCF method (78.37%). This demonstrates the significant advantage of our proposed automatic contrastive prompt extraction approach over the manually designed strategy in CRG.

**Ablation Study of Distillation.** We conduct the following experiments to evaluate different components of the proposed distillation method: (1) Removing the adapter generation network AGN and directly using learnable embeddings as the adapter tokens **A**. (2) Removing $\mathcal{L}_{dis}$ (Eq.4) or $\mathcal{L}_{inp}$ (Eq.5) from the loss functions for distillation. (3) Training $\phi_s$ directly instead of distillation. The performance decline shown in Table 8 demonstrates the critical importance of the proposed AGN, loss functions and distillation method.

visualization results are provided in the Appendix Figure 8.

## 5. Conclusion

This paper introduces CPCF, a novel and effective cross-prompt contrastive framework for referring MLLMs. This framework incorporates a prompt extraction network for automatically generating prompts used in contrastive decoding, a self-training method that leverages unlabeled data for enhanced training, and a distillation approach to reduce the additional computational overhead introduced by contrastive decoding. Extensive evaluation results on multiple benchmarks demonstrate the superiority of our framework over existing methods. We believe this work can provide valuable insights for developing more robust, efficient, and high-performing referring MLLM methods.

## Acknowledgment

T. C. acknowledges support from Dream Set Off - Kunpeng & Ascend Seed Program.

## Impact Statement

This paper presents work whose goal is to advance the field of Machine Learning. There are many potential societal consequences of our work, none which we feel must be specifically highlighted here.

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

# A. More Details of Method

## A.1. Extension to Multi-visual-prompt Scenarios

In the main paper, to simplify the illustration, we describe our method for the inputs containing only one visual prompt $r$. Our method can be easily adapted to multi-$r$ scenarios with the following extensions:

**Contrastive Decoding Prediction.** Given multiple input visual prompts $\{r_i\}_{i=1}^{N_r}$, we generate contrastive prompts $\{\hat{r}_{i,n}\}_{n=1}^{N_{\hat{r}}}$ for each $r_i$ using the method described in Sec.3.2. During the MLLM's contrastive-decoding-based prediction process, we sequentially replace each $r_i$ with each of its corresponding $\hat{r}_{i,n}$ while keeping the other visual prompts unchanged, generating a prediction from the MLLM and contrasting it with MLLM's original prediction from $\{r_i\}_{i=1}^{N_r}$. Finally, the contrastive results obtained from all $\{\{\hat{r}_{i,n}\}_{n=1}^{N_{\hat{r}}}\}_{i=1}^{N_r}$ are averaged to produce the final output.

**RAG in Self-training.** In the self-training method introduced in Sec.3.3, we employ a RAG strategy to retrieve a similar $D^i = \{I^i, r^i, q^i\}$ from $\mathcal{D}$ to guide question generation for each $\{\tilde{I}, \tilde{r}\}$ in $\tilde{\mathcal{D}}$. During this process, we compute an average CLIP feature of region $r^i$ as a local representation $f_l^i$ and that of $\tilde{r}$ as $\tilde{f}_l$, then calculate the cosine similarity $s_l^i$ between each $f_l^i$ and $\tilde{f}_l$ for retrieval. In the actual implementation, to enhance the model's capability in multi-$r$ scenarios, for each $\hat{I}$ in a subset of $\tilde{\mathcal{D}}$, we randomly sample $N_r$ ($N_r > 1$) bounding boxes and convert them into input visual prompts $\{\tilde{r}_j\}_{j=1}^{N_r}$. We compute $\tilde{f}_l^j$ for each $\tilde{r}_j$. For each sample $D_i$ in $\mathcal{D}$ that contains $N_r$ visual prompts $\{r_{i,j}\}_{j=1}^{N_r}$, we also compute the corresponding $\{f_l^{i,j}\}_{j=1}^{N_r}$. Next, we match $\{\tilde{f}_l^j\}_{j=1}^{N_r}$ with $\{f_l^{i,j}\}_{j=1}^{N_r}$ using bipartite matching and compute the cosine similarity for each matched pair. The average similarity over all pairs serves as the local score $s_l^i$, which is used for retrieval in the multi-$r$ scenarios.

**Adapter Module for Distillation.** As illustrated in Sec.3.4, in the single-execution model $\phi_s$, we use an adapter generation network (AGN) to generate a set of adapter modules $\mathbf{A}$ for distillation. When the input contains multiple visual prompts $\{r_i\}_{i=1}^{N_r}$, we also apply Eq.3 to generate an $\mathbf{E}_{i,n}$ for each contrastive prompt $\hat{r}_{i,n}$ of every $r_i$. The resulting $\{\{\mathbf{E}_{i,n}\}_{n=1}^{N_{\hat{r}}}\}_{i=1}^{N_r}$ are then summed and passed through linear layers $W$ to obtain $\mathbf{A}$. During this process, in addition to its original inputs $[\mathbf{f}_r; \mathbf{f}_{\hat{r}_{i,n}}; \mathbf{e}]$, the AGN also receives an additional embedding computed as the position embedding of $\{r_i\}$'s index $i$. This embedding is concatenated with $[\mathbf{f}_r; \mathbf{f}_{\hat{r}_{i,n}}; \mathbf{e}]$ to serve as the AGN's input, providing explicit information about which $\{r_i\}_{i=1}^{N_r}$ the $\hat{r}_{i,n}$ corresponds to.

## A.2. MLLM Prompt for Description Generation

As illustrated in Sec.3.3 of the main paper, we employ the RAG method to generate a question for each image $\hat{I}$ in $\hat{\mathcal{D}}$. To achieve this, we retrieve an image $I_i$ and its visual prompt $r_i$ from the original dataset $\mathcal{D}$. Subsequently, the MLLM is prompted to generate textual descriptions $d_I$ and $d_r$, which correspond to $I_i$ and the region of $r_i$, respectively, to guide the question generation process. The detailed prompts for generating $d_I$ and $d_r$ are as follows:

- **Prompt for generating $d_I$ from $I_i$:** *Describe this image in detail.*
- **Prompt for generating $d_r$ from $r_i$:** *Please provide a description of the region <prompt $r_i$> in a sentence.*

## A.3. Answer Generation with Chain-of-thought

As detailed in Sec.3.3 of the main paper, we use a chain-of-thought (CoT) prompt $p_c$ to generate a more accurate answer $\hat{a}_{cot}$ for each image-prompt pair $\{\hat{I}, \hat{r}\}$ and question $q$, i.e., $\text{MLLM}(\tilde{I}, \tilde{r}, \tilde{q}, p_c)$. The prompt $p_c$ is written as: *Please think step-by-step to answer the question. First, describe the image and indicated region. Next, explain what information is needed to answer the question. Then, identify this information from the image or world knowledge. Finally, based on this reasoning, provide the answer.*

We use GPT-4o to evaluate whether $\hat{a}_{cot}$ or $\hat{a}$ directly generated without using $p_c$ is more accurate. Among 2000 randomly sampled instances $\{\hat{I}, \hat{r}, \hat{q}\}$, GPT-4o determines that $\hat{a}_{cot}$ is better in 91% of the cases. This result validates the rationale of using $\{\hat{a}, \hat{a}_{cot}\}$ pairs for DPO training.

# B. More Details of Tasks and Benchmarks

## B.1. Input Instruction for Evaluation

As illustrated in Sec.4.2 of the main paper, we evaluate the model's performance on four tasks and benchmarks. The input instructions for the referring reasoning and referring captioning tasks on Ferret-Bench are directly included in the dataset.

Table 9. Evaluation of errors caused by **adjacent regions**.

| Models | Accuracy |
|---|---|
| Kosmos2-1.6B (Peng et al., 2023) | 51.5 |
| Shikra-7B (Chen et al., 2023) | 66.8 |
| CogVLM-17B (Wang et al., 2023) | 67.4 |
| Osprey-7B (Yuan et al., 2024) | 73.1 |
| GLaMM-7B (Rasheed et al., 2024) | 74.8 |
| Ferret-7B (You et al., 2023) | 73.9 |
| Ferret-13B (You et al., 2023) | 76.2 |
| **CPCF-7B (Ours)** | **91.3** |
| **CPCF-13B (Ours)** | **92.0** |

Table 10. Evaluation of errors caused by **similar regions**.

| Models | Accuracy |
|---|---|
| Kosmos2-1.6B (Peng et al., 2023) | 39.6 |
| Shikra-7B (Chen et al., 2023) | 52.3 |
| CogVLM-17B (Wang et al., 2023) | 58.8 |
| Osprey-7B (Yuan et al., 2024) | 67.8 |
| GLaMM-7B (Rasheed et al., 2024) | 62.5 |
| Ferret-7B (You et al., 2023) | 67.0 |
| Ferret-13B (You et al., 2023) | 70.9 |
| **CPCF-7B (Ours)** | **83.9** |
| **CPCF-13B (Ours)** | **85.7** |

The MLLM's input instructions for the other three tasks and benchmarks are as follows:

- **Referring Object Classification (ROC):** *Is the object `<prompt r>` a `<class A>` or a `<class B>`?*

- **Referring Text Classification (RTC):** *Is the text `<prompt r>` of the image '`<Text A>`' or `<Text B>`'? please select only one.*

- **Referring Description (RD):** *Can you provide a description of the region `<prompt r>` in a sentence?*

### B.2. Evaluation Metrics

The referring object classification (ROC) task and the referring text classification (RTC) task are both constructed as binary classification tasks. Therefore, we directly use accuracy as the metric to evaluate model performance. For the referring description (RD) task, we follow previous methods (Wu et al., 2024) and adopt four metrics: BLEU@4 (B@4) (Papineni et al., 2002), METEOR (M) (Banerjee & Lavie, 2005), CIDEr-D (C) (Vedantam et al., 2015), and SPICE (S) (Anderson et al., 2016). For the referring reasoning and referring captioning tasks on Ferret-Bench, we use the same approach as Ferret (You et al., 2023), prompting GPT-4 for scoring. Readers can refer to (You et al., 2023) for more details on the evaluation methods used for Ferret-Bench.

# C. More Experimental Results

## C.1. Evaluation of RAG-Generated Questions

As illustrated in Sec.3.3 of the main paper, for each image-prompt pair $\{\tilde{I}, \tilde{r}\}$ in $\tilde{\mathcal{D}}$, we employ the RAG method to retrieve a similar $\{I, r\}$ from $\mathcal{D}$ and use the corresponding question to guide generation of questions for $\{\tilde{I}, \tilde{r}\}$. This approach is motivated by our observation that directly prompting the MLLM to generate questions often results in errors and hallucinations, producing questions unrelated to $\{\tilde{I}, \tilde{r}\}$. In contrast, providing the MLLM with a similar example from $\mathcal{D}$ for reference can significantly improve reliability. To validate this, we use GPT-4o to score the questions generated with and without RAG on a scale from 0 to 10 according to their relevance to the given $\{\tilde{I}, \tilde{r}\}$. Among 2000 randomly sampled $\{\tilde{I}, \tilde{r}\}$ from $\tilde{\mathcal{D}}$, questions generated using RAG achieve an average score of 9.3, significantly higher than the 6.4 achieved without RAG. This demonstrates the rationality of our RAG-based approach.

## C.2. Errors Caused by Misleading Regions

As indicated in the main paper, referring MLLMs often exhibit suboptimal performance due to the confusion caused by misleading areas adjacent to or similar to the target region. To demonstrate the effectiveness of our method in addressing this issue, we conduct the following validation experiments.

**Errors Caused by Adjacent Regions.** To evaluate the error rate of the model being misled by areas adjacent to the indicated visual prompt, we sample 2000 images from the LVIS dataset. For each image, we select two adjacent but category-different bounding boxes, $b_1$ and $b_2$, belonging to classes $c_1$ and $c_2$, respectively. Using $b_1$ as the visual prompt, we ask the referring MLLM "*Is the object `<prompt b_1>` a `<class c_1>` or a `<class c_2>`?*", and then calculate the accuracy based on its responses. As shown in Table 9, our method achieves the best performance, significantly outperforming the baseline method Ferret and all other advanced methods used for comparison. This demonstrates the high effectiveness of our approach in mitigating errors caused by regions adjacent to the input visual prompt.

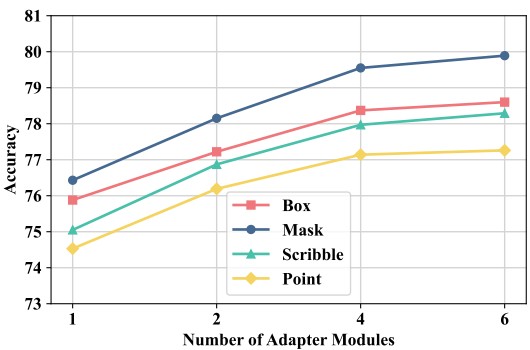

*Figure 6.* Number of adapter modules.

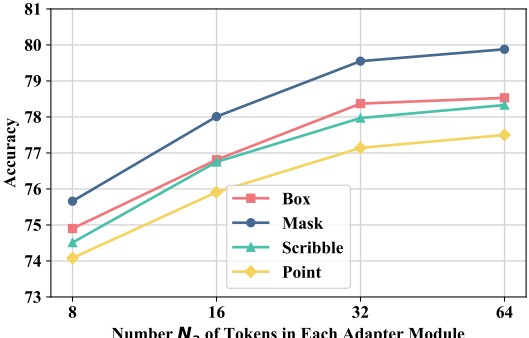

*Figure 7.* Number $N_a$ of tokens in each adapter module.

*Table 11.* Number of queries input to the prompt extraction network.

| Query Number | ROC-Box | ROC-Mask |
|---|---|---|
| 16 | 76.68 | 77.09 |
| 32 | 77.75 | 78.43 |
| 64 | 78.37 | 79.55 |
| 128 | 78.50 | 79.98 |

*Table 12.* Ablation study results of $\alpha$ used in Eq.1.

| $\alpha$ | ROC-Box | ROC-Mask |
|---|---|---|
| 0.1 | 77.91 | 79.24 |
| 0.5 | 78.37 | 79.55 |
| 1 | 78.31 | 79.53 |

*Table 13.* Comparison results with general MLLMs on Ferret-Bench.

| Method | Referring Captioning | Referring Reasoning |
|---|---|---|
| InternVL2.5 | 80.4 | 79.2 |
| Qwen2.5-VL | 79.9 | 80.5 |
| CPCF (Ours) | 83.5 | 82.9 |

**Errors Caused by Similar Regions.** To evaluate the error rate caused by the areas similar to the input visual prompt, we sample 2000 images from the LVIS dataset, each containing two bounding boxes, $b_1$ and $b_2$, belonging to the same category $c$. We use GPT-4o to generate descriptions $t_1$ and $t_2$ respectively highlighting a different attribute between $b_1$ and $b_2$ (e.g., color, pose, position, and relationships with surrounding objects). After that, we ask the referring LLM "*Which text correctly describe the properties of object <prompt $b_i$>, '<description $t_1$>' or '<description $t_2$>'? Please select only one.*" The results presented in Table 10 highlight the significant superiority of our CPCF compared to previous methods, demonstrating the effectiveness of our approach in avoiding errors caused by misleading areas similar to the visual prompt.

### C.3. Ablation Study of Hyperparameters

**Number of Adapter Modules.** In our method, to facilitate training during the distillation process, $N$ adapter modules are inserted at evenly distributed intervals throughout the MLLM of the single-execution model $\phi_s$. We conduct an ablation study on the number $N$ of adapter modules, with the results presented in Figure 6. When $N < 4$, the model's performance improves as the number of adapters increases. However, when $N > 4$, the performance saturates, and further increasing the number of adapters yields only minimal improvement. Based on these findings, we use 4 adapter modules in our method.

**Number $N_a$ of Tokens in Each Adapter Module.** We further evaluate the effect for the number $N_a$ of tokens in each adapter module. As shown in Figure 7, a similar trend is observed: when $N_a < 32$, increasing the number of tokens significantly improves the model's performance; however, when $N_a > 32$, the performance tends to saturate. Based on these observations, we set the number of tokens in each adapter module to 32.

**Number of Learnable Queries Input to the Prompt Extraction Network.** In our proposed prompt extraction network, a set of learnable query embeddings are input into the image encoder. As shown in Table 11, when the number of such learnable queries is too small, the network may miss critical information, leading to performance degradation; whereas when the number is too large, the model performance saturates, resulting in limited improvement if further increasing the query number (only +0.17 accuracy on ROC-Box when the query number increases from 64 to 128). Based on these results, we choose 64 as the number of input query embeddings in our method.

**Ablation Study of $\alpha$ in Eq.1.** As shown in the Table 12, when $\alpha$ is set to 0.1, 0.5, or 1, the model achieves ROC-box accuracy of 77.91, 78.37, and 78.31, respectively, demonstrating stable performance across a range of values and consistently outperforming the previous SOTA result (72.83). In fact, the setting of $\alpha$ is directly adopted from existing contrastive decoding approaches, and we found that it can already work very well without requiring specific modifications.

## C.4. Comparison with General MLLMs

Some recent general MLLMs such as Qwen and InternVL also supports referring tasks. We further compare our CPCF with these MLLMs and the results on Ferret-Bench are presented in Table 13. As shown by the table, our CPCF-7B outperforms both InternVL2.5-8B and Qwen2.5-VL-7B significantly, demonstrating the advantages and effectiveness of our novel method benefiting the novel designs tailored to the referring tasks proposed in this paper.

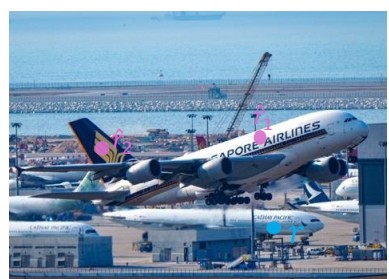

**Question:** Which airline does the aircraft in [region] belong to?

**Ferret:** The aircraft belongs to Singapore Airlines.

**Ours:** It belongs to Cathay Pacific.

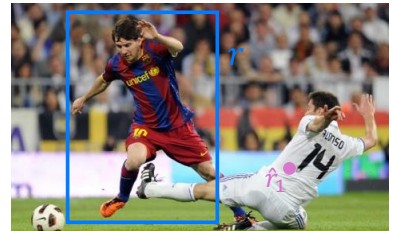

**Question:** What is this player [region] doing?

**Ferret:** This player is attempting to steal the ball from another player.

**Ours:** This player is dribbling past another player's defense.

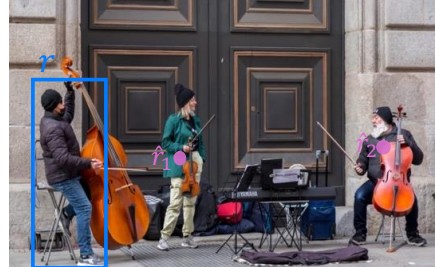

**Question:** What is the role of the man [region] in the band?

**Ferret:** He is responsible for playing the viola.

**Ours:** He is the cellist in the band.

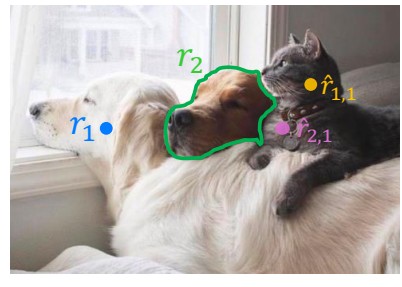

**Question:** What is the difference between [region1] and [region2]?

**Ferret:** [region1] is a dog while [region2] is a cat.

**Ours:** Their fur colors are different. [region1] is white, and [region2] is yellow.

*Figure 8.* More visualization results of the contrastive prompts generated by the prompt extraction network and the quantitative comparison between our method and Ferret. The incorrect contents generated by Ferret and the corresponding correct contents generated by our method are highlighted in red and green, respectively.

