# OpenReview forum: "CPCF: A Cross-Prompt Contrastive Framework for Referring Multimodal Large Language Models"
_ICML.cc/2025/Conference — ICML 2025 poster_

### Official Review · Reviewer_WGMH · 2025-03-12

**Overall Recommendation:** 3

**Summary:**

This paper focus on addressing the incorrect responses of referring MLLMs tailored to misleading areas adjacent to or similar to the target region. Specifically, it introduces contrastive visual prompts generated by prompt extraction network, which is trained on extra dataset. Besides, to alleviate the computational cost, the distillation is adopted to run single-execution decoding. The writing is good in general. The experimental results shows its effectiveness of proposed method.

However, training on extra dataset compared with other competitors weakens its effectiveness. Besides, the training process is much complex compared with competitors.

**Claims And Evidence:**

Yes, the claims are supported by quantitative and qualitative results.

**Essential References Not Discussed:**

The references seem adequate.

**Experimental Designs Or Analyses:**

Yes, the comparison and ablation studies show its effectiveness of proposed method.

**Methods And Evaluation Criteria:**

Yes, the method seems make sense for the problem.

**Other Comments Or Suggestions:**

The ‘[region]’ on the top part of figure 3 should be green.

**Other Strengths And Weaknesses:**

(1)The training process is much complex than competitors, which consumes computation resources.
(2)Besides, it utilizes extra dataset to train prompt extraction network and also the MLLM. In Line 315-316, the definition ‘not used’ and ‘included’ is measured from image-level? The region-level information may be leaked to test set.

**Questions For Authors:**

The contrast visual prompts are point or mask? Is it fixed across all datasets?

**Relation To Broader Scientific Literature:**

It uses Ferret as baseline to conduct referring.

**Theoretical Claims:**

There are no proofs for theoretical claims.

---

> ### Author Rebuttal · Authors · 2025-03-31
>
> ### **Q1 Training process**
>
> Yes, we agree that our training process involves additional stages—specifically, self-training and distillation—which require 30 hours and 25 hours, respectively, to train for 50K steps on 8 A100 GPUs for the 7B model. However, we would like to kindly clarify the following points:
>
> * (1) As shown in Table 4 of the main paper, **both stages are crucial**: self-training significantly improves model performance, while distillation greatly enhances computational efficiency. These results validate the rationality of our multi-stage training pipeline.
> * (2) Moreover, **if training resources are limited, it is entirely feasible to largely reduce the number of training steps while still maintaining strong performance**. We observe that the model can achieve over 95% of its final performance after just 25K steps, reducing the required training time to 15 hours and 12 hours for the two stages, respectively. Even under this shorter training schedule, the model still achieves SOTA performance, making the overall training cost and effectiveness entirely acceptable for practical use.
> * (3) Additionally, once training is completed, **the additional computational cost during inference for our method is very minimal—only +4.3% compared to the baseline Ferret**. This efficiency is achieved because our model supports end-to-end inference without requiring multiple MLLM executions as other contrastive decoding (CD) methods,  which substantially reduces CD's computational complexity, highlighting the advantages of our method.
>
> ### **Q2 Extra dataset**
>
> Yes, our method utilizes an additional dataset for training. This is intended to mitigate potential overfitting issues that may arise when performing contrastive decoding training directly on the original dataset (please refer to Lines 192–218 of the main paper for details). However, we would like to respectfully clarify the following points:
>
> *  (1) The additional dataset we use is NOT a fully annotated, supervised dataset, but rather an unannotated corpus without any question-answer labels,  which is **easy to collect in real-world scenarios**. Moreover, training a model with such unlabeled data is non-trivial, and we have carefully designed a self-training approach to address this challenge.  As such, **a new method for training referring MLLMs using unlabeled data is itself a key contribution of this work.**
>
> *  (2) **Even without using the extra dataset, our method can still achieve state-of-the-art performance**. To be specific, using the original dataset $\mathcal{D}$ without the extra dataset $\tilde{\mathcal{D}}$ in our method achieves an accuracy of 74.55 on ROC-Box, which still significantly outperforms previous approaches such as Ferret (71.71), Osprey (72.15), and Shikra (64.60). This indicates that the extra dataset only serves as a strategy for further enhancement, but is not the sole reason for our method’s strong performance. The proposed network architecture and training strategy also play a crucial role in achieving high effectiveness. In the revised version of the paper, we will incorporate the results without the extra dataset into Table 1 and Table 2. Thank you very much!
>
> ### **Q3  "not used’ and "not included’ in Line 315-316**
>
> Here, the definitions ‘not used’ and ‘included’ are measured from both image-level and region-level, thus the region-level information will NOT be leaked to the test set.
>
> ### **Q4 The ‘[region]’ on the top part of figure 3 should be green.**
>
> Thank you for indicating this typo! We will fix it in the revised paper.
>
> ### **Q5 The contrast visual prompts are point or mask? Is it fixed across all datasets?**
>
> The contrastive prompt is a point, and this choice is fixed across all datasets. We also explored generating different types of prompts based on the input type—for example, generating box contrastive prompts for box inputs and mask contrastive prompts for mask inputs. However, this approach increases the optimization difficulty for the prompt extraction network, as it would require generating multiple prompt types from a single network. While using a separate prompt extraction network for each type is a potential solution, it would significantly increase the model’s parameter count. Therefore, we adopt a unified and simplest form of contrastive prompt across all datasets: the point.
>
>
> **In summary, our method can achieve SOTA performance with an acceptable training time and requires very minimal additional computation during inference (only +4.3% than Ferret). While an extra dataset is used, it does not require any manual question–answer annotations, making it easy to obtain in practice.** Thank you again for your valuable commnets and thoughtful questions! We hope that our responses can address your concerns, and we would be happy to engage in any further discussion if you have any questions. Thank you so much!

---

### Official Review · Reviewer_Za66 · 2025-03-14

**Overall Recommendation:** 3

**Summary:**

The paper introduces CPCF (Cross-Prompt Contrastive Framework), a novel approach to improving Referring Multimodal Large Language Models. These models extend standard Multimodal Large Language Models by allowing them to focus on specific image regions using visual prompts. CPCF leverages a contrastive decoding strategy, where responses generated from an intended visual prompt are contrasted against those generated from misleading prompts.

Through extensive experiments across multiple benchmarks, CPCF achieves state-of-the-art performance, demonstrating significant improvements in multiple tasks. The model effectively mitigates errors caused by misleading regions, significantly outperforming previous approaches like Ferret.

**Claims And Evidence:**

The claims made in the paper are generally supported by empirical evidence, including extensive experiments and comparisons with prior models.

However, I have several severe questions. The paper mentions that the frameworks "eliminating the uncertainty and instability inherent in manual methods. " How is it possible that the method "eliminates" uncertainty and instability? And what do these two properties, uncertainty and instability, mean in practice?

**Essential References Not Discussed:**

N/A

**Experimental Designs Or Analyses:**

The contrastive decoding effectiveness is demonstrated through both quantitative results (Tables 1-3) and qualitative visualizations (Figure 5). The authors also considered multiple baselines and datasets for comparison and evaluation. The ablation study results are also sound.

**Methods And Evaluation Criteria:**

The contrastive decoding framework directly addresses the issue of models being misled by adjacent or similar visual regions, and the prompt extraction network provides an automated way to generate meaningful contrastive prompts. I suppose that this method is reasonable.

**Other Comments Or Suggestions:**

The figures and illustrations are slightly messy, especially the framework figure. Eq. (5) is out of box.

**Other Strengths And Weaknesses:**

Strengths:

The self-training mechanism leverages unlabeled images to generate synthetic question-answer pairs, enhancing model robustness without requiring extensive manual annotation.

Weaknesses:

CPCF introduces multiple additional components (contrastive decoding, self-training, and distillation), making training more computationally intensive than standard referring MLLMs. The authors did not provide a sufficient analysis about the computational costs, which could be a problem.

The weighting of contrastive prompts and distillation settings (e.g., hyperparameters like α in contrastive decoding) may significantly impact performance, requiring careful tuning.

**Questions For Authors:**

N/A

**Relation To Broader Scientific Literature:**

The novelty lies in task-specific contrastive decoding, automated prompt selection, and efficiency-aware distillation. These are meaningful contributions to the literature.

**Theoretical Claims:**

There are no theoretical claims provided or any proof. The equations in this paper seem to be correct.

---

> ### Author Rebuttal · Authors · 2025-03-31
>
> Thank you so much for your time in reviewing our paper and the valuable comments! We are sincerely encouraged by your recognition that our contributions are meaningful, the proposed method is reasonable, and the ablation study results are sound. We would like to provide the following responses to address your concerns and questions:
>
> ### **Q1 Clarification about "eliminating the uncertainty and instability inherent in manual methods"**
>
> As indicated in the introduction, we observe that the model tends to generate incorrect answers tailored to misleading regions that are adjacent or similar to the target object. Motivated by this observation, we initially attempted to manually select a random point within these misleading regions as the contrastive prompt for contrastive decoding. However, we empirically found that model performance is highly sensitive to the choice of this point. For instance, selecting two different points within the same misleading region—only 10 pixels apart—may result in significantly different contrastive decoding outcomes. This sensitivity is why we describe manual methods as exhibiting “uncertainty and instability.” In contrast, our prompt extraction network, trained through deep learning optimization, can learn to automatically identify the most appropriate contrastive prompt, thereby alleviating the instability introduced by random manual sampling, leading to better performance. We acknowledge that our original use of the term “eliminate” was too absolute, and we will revise it to “alleviate” in the updated version of the paper. Thank you!
>
> ### **Q2 Computational costs of training**
>
> Thank you for this valuable comment! Building on the pretrained Ferret model, our method introduces two additional training stages: (1) self-training and (2) distillation,  which require 30 hours and 25 hours, respectively, to train for 50K steps on 8 A100 GPUs for the 7B model. In practice, if training resources are limited, it is entirely feasible to largely reduce the number of training steps while still maintaining strong performance. We observe that the model can achieve over 95% of its final performance after just 25K steps, reducing the required training time to 15 hours and 12 hours for the two stages, respectively. Even under this shorter training schedule, the model still achieves SOTA performance, making the overall training cost and effectiveness entirely acceptable for practical use. Additionally, we would like to kindly emphasize that, once training is completed, **the additional computational cost during inference for our method is very minimal—only +4.3% compared to the baseline Ferret**. This efficiency is achieved because our model supports end-to-end inference without requiring multiple MLLM executions as other contrastive decoding (CD) methods,  which substantially reduces CD's computational complexity, highlighting the advantages of our method.
>
> ### **Q3 Hyperparameter**
>
> Thank you for the comment! Our method is not sensitive to the choice of the hyperparameter $\alpha$ in Eq. 1. As shown in the table below, when $\alpha$ is set to 0.1, 0.5, or 1, the model achieves ROC-box accuracy of 77.91, 78.37, and 78.31, respectively—demonstrating stable performance across a range of values and consistently outperforming the previous SOTA result (72.83). In fact, most of the hyperparameters in our method—such as $\alpha$ and other training settings—can be directly adopted from existing contrastive decoding and referring MLLM approaches. We found that these settings already work very well without requiring specific modifications. Nevertheless, we are happy to conduct more ablation studies across a wider range of implementation settings and will include these results in the revised paper.
> | $\alpha$ | ROC-Box | ROC-Mask |
> | -------- | :-----: | :------: |
> | 0.1      |  77.91  |  79.24   |
> | 0.5      |  78.37  |  79.55   |
> | 1        |  78.31  |  79.53   |
>
> ### **Q4 Figure issues**
>
> The out-of-box issue will be fixed. Thank you so much!
>
> Thank you again for your valuable comments and thoughtful questions. We hope that our responses can address your concerns, and we sincerely wish you all the best in your future life and work!

---

### Official Review · Reviewer_Vjf8 · 2025-03-18

**Overall Recommendation:** 4

**Summary:**

The article aims to solve the defects of existing referring MLLMs: it is difficult to accurately locate the prompt, and designs several solutions: 1. an effective referring MLLM framework that contrasts input prompts with contrastive prompts from misleading regions, 2. an automatic prompt extraction mechanism to identify optimal contrastive prompts, a self-training method to improve network optimization

## update after rebuttal
I think the author answered my questions very well. After the author supplemented the experiment, the experimental part of the article became more complete. In addition, I agree with the author's explanation of DPO and SFT. I suggest that the author supplement this content in the revision. I decided to maintain 4 points.

**Claims And Evidence:**

The article has clear experimental results and robustness analysis experiments to prove its credibility

**Essential References Not Discussed:**

No

**Experimental Designs Or Analyses:**

The experiments in this article are quite sufficient, but the comparison objects lack general MLLMs, such as Qwen-VL2.5. However, it has a detailed robustness analysis experimental module.

**Methods And Evaluation Criteria:**

The article proposes three different strategies, including iterative DPO, which can enhance the performance of the model, which is verified in the robustness analysis experiment of the article.

**Other Comments Or Suggestions:**

As shown in Other Strengths And Weaknesses

**Other Strengths And Weaknesses:**

Strengths：
1. The experimental results of the article are very good, far exceeding the previous referring MLLMs
2. The article provides a very detailed robustness analysis experiment, which makes me very confident in the effectiveness of the designed module.
3. The innovation of this article is quite clever. It integrates the idea of ​​contrastive decoding into referring MLLMs and achieves good results.

Weaknesses：
I think the overall quality of the article is high, and there are no logical errors or experimental setting errors, but I have the following doubts about the experimental part, I think it would be better to explain these clearly

1. Why not compare some general MLLMs? For example, Qwen, InternVL, which both support referring tasks?

2. Why use iterative DPO? Have you tried iterative training similar to STAR?

3. Why is there no improvement in the trained model of 13B compared to 7B? Does it mean that this pipeline is difficult to scale up?

**Questions For Authors:**

As shown in Other Strengths And Weaknesses

**Relation To Broader Scientific Literature:**

No

**Theoretical Claims:**

The article is an application-oriented article and does not provide any theoretical proof.

---

> ### Author Rebuttal · Authors · 2025-03-31
>
> Thank you very much for your time in reviewing our paper and the valuable comments! We are sincerely encouraged by your recognition that our experimental results are good, the robustness analysis is detailed, the innovation is clever, and the overall quality of the article is high. For your questions and concerns, we are happy to provide the following responses:
>
> ### **Q1 Comparison with general MLLMs**
>
> Great idea! The comparison results with InternVL2.5-8B and Qwen2.5-VL-7B on Ferret-Bench are shown in the Table below. Our CPCF-7B outperforms both methods significantly, demonstrating the advantages and effectiveness of our novel method. We will compare on more benchmarks and incorporate these results into the revised paper.
>
> | Method          | Referring Cpationing | Referring Reasoning |
> | --------------- | :------------------: | :-----------------: |
> | InternVL2.5     |         80.4         |        79.2         |
> | Qwen2.5-VL      |         79.9         |        80.5         |
> | **CPCF (Ours)** |       **83.5**       |      **82.9**       |
>
>
> ### **Q2 Usage of DPO and STAR**
>
> Our method employs DPO rather than the standard SFT-based iterative training like STaR. This choice is motivated by the fact that compared to SFT, preference-based fine-tuning with DPO allows for more fine-grained control over model optimization and has been shown in prior work to better mitigate hallucination issues. The results in Table 6 of the main paper demonstrate the effectiveness of DPO: compared to conventional CE loss for SFT, DPO achieves improvements of 2.90, 3.47, 2.75, and 2.13 on box, mask, scribble, and point prompt types, respectively. Nevertheless, we agree that the STaR is a promising approach, and we plan to explore its potential in the context of referring MLLMs in future work. Thank you so much!
>
> ### **Q3 Improvement of 13B compared to 7B**
>
> The limited performance improvement observed when scaling from the 7B to the 13B model may result from the relatively small size of existing referring multimodal datasets, which are insufficient to exploit the full potential of larger models. In fact, prior SOTA methods have also similarly exhibited marginal performance differences between their 7B and 13B variants; for example, Ferret-v2-13B even underperforms its smaller version, Ferret-v2-7B, on the Ferret-Bench benchmark. In contrast, despite a similarly modest performance gap, **our CPCF-13B consistently outperforms CPCF-7B across all benchmarks**, demonstrating our method’s effective scalability to larger models. For example, on the ROC task, CPCF-13B improves over CPCF-7B by 0.72, 1.17, 0.88, and 1.05, respectively across the 4 input prompt types. We believe that the development of larger and higher-quality referring multimodal datasets is crucial for unlocking the full potential of large referring MLLMs, which we identify as a key direction for our future work.
>
>
>
> Thank you again for your valuable comments. We hope that our responses can address your concerns, and we sincerely wish you all the best in your future life and work!

---

### Official Review · Reviewer_nWKK · 2025-03-19

**Overall Recommendation:** 3

**Summary:**

This paper introduces CPCF, a cross-prompt contrastive framework for referring multimodal large language models. It improves region-specific response accuracy by automatically generating contrastive prompts from misleading regions, training these through a self-training strategy on additional unlabeled data, and then distilling the multi-execution model into a single-pass efficient model—all validated via extensive experiments across several referring benchmarks.

**Claims And Evidence:**

Contrastive decoding using automatically extracted prompts reduces errors from distracting regions.\
Self-training on extra data (with a RAG-based question generation method) further improves prompt extraction. \
The distillation method can lower inference costs while preserving performance.

**Essential References Not Discussed:**

No

**Experimental Designs Or Analyses:**

Yes

**Methods And Evaluation Criteria:**

Please check the section **Other Strengths And Weaknesses**.

**Other Comments Or Suggestions:**

Please check the section **Other Strengths And Weaknesses**.

**Other Strengths And Weaknesses:**

1. In Section 3.2, the authors construct a semantic similarity map and a relative distance map, process them through a lightweight CNN, and incorporate them into the image tokens. Directly adding these maps to the image tokens may introduce feature contradictions, leading to noisy updates.
2. The quality of $\tilde{D}$ must be exceptionally high, requiring both $D$ and $\tilde{D}$ from the same image domain to ensure consistency.
3. Although the self-training process in Section 3.3 leverages RAG and DPO, each component remains susceptible to errors, and the computational cost is significantly high.
4. The distillation step reduces inference costs, but the algorithm in general increases the overall complexity and computational burden of training. I recommend including experimental results comparing inference time between the proposed method and baselines to better illustrate efficiency gains.

**Questions For Authors:**

N/A

**Relation To Broader Scientific Literature:**

The paper builds upon and extends several lines of recent research. It relates to works on referring MLLMs (e.g., Ferret, Kosmos2), contrastive decoding for reducing hallucinations, and self-training and distillation techniques common in large-model training.

**Theoretical Claims:**

The paper does not include theoretical claims.

---

> ### Author Rebuttal · Authors · 2025-03-31
>
> We sincerely thank you for the time and effort you dedicated to reviewing our paper, as well as for your highly valuable comments!  We would like to provide the following responses to address your concerns and questions:
> ### **Q1 Concerns on feature contradictions**
> Thank you for this comment! We would like to kindly clarify that the semantic similarity map and relative distance map are NOT directly added to the image tokens. Instead, they are first processed by a learnable CNN before being fused with the image tokens. Through training, the CNN can be optimized to transform the map information into the same feature space as the image tokens. Moreover, the features generated by the CNN have the same spatial sizes as the image tokens produced by the vision encoder. Therefore, the problem of feature contradictions will not arise. In fact, this is a common strategy also widely used in many other models—for example, in the prompt encoder of the Segment Anything Model.
> ### **Q2&Q3 Quality of $\tilde{\mathcal{D}}$ and errors / computations of RAG and DPO**
> Thank you for raising this important point! We would like to kindly provide the following clarifications:
> * (1) **The generated data (including questions and answers) in $\tilde{\mathcal{D}}$ are of high quality, thus they can be effectively used for training without introducing many errors**.  As indicated in Appendix C.1, we use GPT-4o to score the questions generated with RAG on a scale from 0 to 10 according to their relevance to the given image and input prompt. Among 2000 randomly sampled data from $\tilde{\mathcal{D}}$, these generated questions achieve a very high average score of 9.3, demonstrating their excellent quality. Furthermore, as reported in Appendix A.3, GPT-4o judges that in 91% of the sampled cases, the CoT-based answers are more accurate than the standard answers, demonstrating the effectiveness of using these pairs for DPO training. We also conduct a user study on 200 randomly selected generated samples, where volunteer ratings indicate that 94% of the generated data are correct and free of significant errors. These results demonstrate the robustness and effectiveness of our RAG- and CoT-based data generation method.
> * (2)  **Even with a small amount of errors, the impact on model performance is very minimal**. We find that models trained on our generated data perform almost identically to those trained on fully clean data. Specifically, we use GPT-4o to filter out noisy question–answer pairs from the generated dataset $\tilde{\mathcal{D}}$—approximately 8% of the total—and replace them with corrected versions. The model retrained on this cleaned dataset achieves ROC-Box and ROC-Mask accuracies of 78.90 and 79.96, respectively—only marginally higher than those of our original model (78.37 and 79.55), with differences of just 0.53 and 0.41. These minor performance gaps indicate that small errors in the generated cannot affect model performance significantly, further demonstrating the robustness of our method.
> * (3) The original dataset $\mathcal{D}$ is highly diverse, containing rich images from a wide range of domains. Therefore, when constructing $\tilde{\mathcal{D}}$, it is unnecessary to deliberately search for images that specifically match the domains in $\mathcal{D}$. Instead, randomly sampling images from various domains is sufficient and can be easily implemented in practical applications.
> * (4) Compared to directly using $\mathcal{D}$ or training with conventional CE loss, **using RAG to generate dataset $\tilde{\mathcal{D}}$ or training with the DPO loss increases training time by only 4.7 hours and 9.2 hours**, respectively—an overhead we believe is entirely acceptable in practical applications, especially considering the significant performance improvements they bring (as shown in main paper Table 6).
> ### **Q4 Comparison of inference time between the proposed method and baselines**
> Good suggestion! The comparison of inference time between our method and the baseline Ferret is presented in Section 4.4, Lines 429–432. **Compared to Ferret, our method incurs only a 4.3% increase in average inference time while achieving significantly better performance** (please refer to Table 1). Additionally, Table 4 of the main paper compares the inference time of our method with and without distillation, showing that the distilled model reduces inference time by more than threefold. We further compare our CPCF with Shikra and GLaMM, and CPCF requires only an additional 8.9% and 2.5% average inference time, respectively. These demonstrate that the additional inference computation introduced by our method is minimal. We will include a comprehensive inference time comparison with all methods listed in Table 1 and incorporate the results into the revised paper. Thanks!
>
> Thank you again for your valuable comments. We hope that our responses can address your concerns, and we sincerely wish you all the best in your future life and work!

---

### Official Review · Reviewer_Xoy3 · 2025-03-19

**Overall Recommendation:** 4

**Summary:**

This paper addresses the performance limitations of referring multimodal large language models (MLLMs), which often misinterpret ambiguous or misleading visual regions during referring comprehension tasks. To overcome this limitation, the authors propose the Cross-Prompt Contrastive Framework (CPCF), which improves both region understanding and response generation through systematic contrastive prompting.

The framework operates through three core innovations: (i) it automatically extracts discriminative visual prompts using a Q-Former module to highlight critical image regions; (ii) it employs a self-training paradigm that combines retrieval-augmented question generation with direct preference optimization (DPO) to enhance response quality; and (iii) it incorporates a knowledge distillation mechanism that facilitates efficient contrastive decoding while significantly reducing computational overhead.

Furthermore, this work presents an integration of retrieval-augmented generation for the creation of synthetic training data, along with contrastive learning principles tailored for visual-language alignment. Evaluations across multiple benchmarks demonstrate that CPCF achieves state-of-the-art performance, with quantitative results indicating significant accuracy improvements over existing methods. The authors further validate that the contrastive approach of their framework effectively reduces hallucination errors while maintaining computational efficiency through their proposed distillation strategy.

**Claims And Evidence:**

Yes, the claims are supported by evidence in the paper.

**Essential References Not Discussed:**

N/A.

**Experimental Designs Or Analyses:**

Yes.

**Methods And Evaluation Criteria:**

Yes, appropriate benchmarks and metrics are used.

**Other Comments Or Suggestions:**

N/A.

**Other Strengths And Weaknesses:**

### Strengths
- **Novel Methodology**: CPCF is a novel framework that leverages contrastive prompts to enhance referring MLLMs. The proposed framework is well-developed and incorporates several improvements that enhance both accuracy and efficiency.
- **Extensive Experimental Validation**: The authors conduct extensive experiments across multiple benchmarks and four diverse tasks, demonstrating state-of-the-art performance.
- **Good Writing**: The paper is clearly written and well-structured.

### Weaknesses
- **Lack of inference speed comparison**: The paper asserts that the proposed distillation technique enhances efficiency; however, it lacks a direct comparison with other methods regarding inference time.
- **Missing ablation on contrastive prompt quantity**: Table 5 presents ablation studies on the semantic similarity map and the relative distance map; however, there is no analysis of the impact of the number of contrastive prompts used.
- **Missing baselines**: Table 7 demonstrates that directly fine-tuning Ferret on the new dataset $\hat{D}$ achieves a box accuracy of 74.22, while CPCF with contrastive decoding attains an accuracy of 78.37. However, there has been no evaluation of Ferret-7B fine-tuned using contrastive decoding methods (e.g., CRG).

**Questions For Authors:**

Refer to the weaknesses above.

**Relation To Broader Scientific Literature:**

N/A.

**Theoretical Claims:**

The paper does not seem to make complex theoretical claims requiring proof verification. However, the core idea of contrastive decoding  is supported by empirical results.

---

> ### Author Rebuttal · Authors · 2025-03-31
>
> Thank you very much for your time in reviewing our paper and the valuable comments! We are sincerely encouraged by your recognition that our method is novel and well-developed, experiments are extensive, and writing is good. For your questions and concerns, we are happy to provide the following responses:
>
> ### **Q1 Inference speed comparison**
>
> Thank you for this comment! The comparison of inference time between our method and the baseline Ferret is presented in Section 4.4, Lines 429–432. **Compared to Ferret, our method incurs only a 4.3% increase in average inference time while achieving significantly better performance** (please refer to Table 1). **We further compare our CPCF with Shikra and GLaMM: CPCF requires only 8.9% and 2.5% additional inference time, respectively, while achieving accuracy improvements of 13.77 and 8.44.** This demonstrates that the additional inference computation introduced by our method is minimal while the performance advantage is significant. We will include a comprehensive inference time comparison with all methods listed in Table 1 and incorporate the results into the revised paper.
>
> ### **Q2 Ablation on contrastive prompt quantity**
>
> Thank you for your comment. We would like to clarify that **the number of contrastive prompts is NOT a predefined or fixed hyperparameter**; instead, it is automatically determined by the network. Specifically, as detailed in Lines 184–189 of the main paper, we utilize the Mean Shift algorithm to cluster the positions of all valid prompts to generate the contrastive prompts. Since Mean Shift is inherently a clustering method without a fixed cluster number, the number of resulting contrastive prompts varies accordingly, with an average of 3.2 in our experiments. We will provide a more detailed statistical analysis in the revised paper. A potentially related hyperparameter is **the number of learnable queries input to the Prompt Extraction Network**. As shown in the table below, when the number of learnable queries is too small, the network may miss critical information, leading to performance degradation; whereas when the number is too large, the model performance saturates, resulting in limited improvement if further increasing the query number (only +0.17 accuracy on ROC-Box when the query number increases from 64 to 128). We will include a more detailed ablation study on the number of learnable queries in the revised paper. Thanks!
>
> | Query Number | ROC-Box | ROC-Mask |
> | ------------ | :-----: | :------: |
> | 16           |  76.68  |  77.09   |
> | 32           |  77.75  |  78.43   |
> | 64           |  78.37  |  79.55   |
> | 128          |  78.50  |  79.98   |
>
>
>
> ### **Q3 Ferret-7B finetuned using CRG**
>
> Good advice! We follow your suggestion by fine-tuning Ferret-7B on $\tilde{\mathcal{D}}$ using the contrastive decoding method CRG. As shown in the table below, **this method achieves an accuracy of 75.95% on the ROC-Box scenario, outperforming baseline Ferret (71.71%) and naive fine-tuning (74.22%), but still falling significantly short of our CPCF method (78.37%)**. This demonstrates the significant advantage of our proposed automatic contrastive prompt extraction approach over the manually designed strategy in CRG. We will include this result into Table 7 of the revised paper.
>
> | Method                         |  ROC-Box  | ROC-Mask  |
> | :----------------------------- | :-------: | :-------: |
> | Ferret                         |   71.71   |   72.39   |
> | Ferret + naive fine-tuning     |   74.22   |   74.37   |
> | Ferret + CRG (w/o fine-tuning) |   75.40   |   76.20   |
> | Ferret + CRG (w/ fine-tuning)  |   75.95   |   77.06   |
> | **CPCF (Ours)**                | **78.37** | **79.55** |
>
>
>
> Thank you again for your valuable comments and thoughtful questions! We hope that our responses can address your concerns, and we would be happy to engage in any further discussion if you have any questions. Thank you so much!

---

### Official Review · Reviewer_avef · 2025-03-20

**Overall Recommendation:** 2

**Summary:**

This paper presents CPCF, a cross-prompt contrastive learning framework designed to enhance the performance of referring multimodal large language models (MLLMs). The proposed approach aims to address a key issue in existing referring MLLMs: errors caused by misleading visual regions adjacent to or similar to the target region. The authors introduce three contributions:
1. A prompt extraction network to automatically identify optimal contrastive prompts.
2. A self-training method leveraging unlabeled data to improve training quality.
3. A distillation approach to reduce the computational overhead associated with contrastive decoding.

The paper is well-organized, the motivation is clearly stated, and experiments demonstrate the effectiveness of CPCF across multiple benchmarks. However, there are some areas that require improvement, as outlined below.


**Pros:**
1. The proposed method is thoroughly evaluated across multiple datasets, including Referring Object Classification (ROC), Referring Text Classification (RTC), and Referring Description (RD).
2. The results show state-of-the-art performance, significantly outperforming existing models like Ferret, GPT4RoI, and Shikra.

**Cons:**

1. There are many shortcomings in the writing of the paper, including some grammatical issues. For example, in line 022, the phrase "from...from...from..." appears. There are also other expression issues throughout the paper. I'm also confused by the naming of the method (CPCF) in this paper.
2. Figure 1 does not reflect the image editing experiment mentioned on the right side of line 39 in the paper. It is unclear what the authors intend to convey with Figure 1 in the main text and Figure 8 in the supplementary material.
3. Grounding or referring multimodal large language models is currently a very popular topic, but the authors' introduction to related work is incomplete. They may refer to relevant works mentioned in the recent survey "Towards Visual Grounding: A Survey" [1]. For example, besides Ferret and Shikra, which are mentioned in the paper, there are many similar works, such as LLaVA-Grounding [2], Grounding-GPT[3], Next-Chat, Groma, LION, Ferret-V2, and u-LLaVA, etc, most of these related works should be discussed.
4. The self-training approach using generated data mentioned in this paper is not novel. Similar methods have been widely used in weakly or unsupervised visual grounding works, such as CLIP-VG [4] and VG-annotator [5]. However, the author without include any discussions.
5. While the paper claims CPCF is efficient, a more detailed breakdown of computational overhead (e.g., FLOPs, memory usage, or inference speed comparisons) would strengthen this claim.
6. The paper compares CPCF with Ferret + VCD, Ferret + ICD, and Ferret + CRG in Table 3. However, it lacks a qualitative discussion on why CPCF outperforms these methods beyond numerical results.
7. A deeper comparison with CRG (Wan et al., 2025) is needed, as both use contrastive decoding but differ in implementation. The authors should highlight how CPCF's automatic prompt extraction provides advantages over CRG's perturbation-based method.
8. Table 4 shows an increase in inference time without distillation. The authors should clarify whether the distillation affects model accuracy trade-offs.
9. Most importantly, the experiments in this paper are solid, but the work appears to be an combination of existing modules and methods. Techniques such as prompting, pseudo-label self-training, and distillation are commonly used in current research. The paper does not present particularly outstanding innovations that would be sufficient for publication at a top-tier conference like ICML.

--


[1] Towards Visual Grounding: A Survey. arXiv preprint arXiv:2412.20206.

[2] Llava-grounding: Grounded visual chat with large multimodal models. In European Conference on Computer Vision (pp. 19-35). Cham: Springer Nature Switzerland.

[3] GroundingGPT: Language Enhanced Multi-modal Grounding Model. In Proceedings of the 62nd Annual Meeting of the Association for Computational Linguistics (Volume 1: Long Papers) (pp. 6657-6678).

[4] VG-Annotator: Vision-Language Models as Query Annotators for Unsupervised Visual Grounding. In 2024 IEEE International Conference on Multimedia and Expo (ICME) (pp. 1-6). IEEE.

[5] Clip-vg: Self-paced curriculum adapting of clip for visual grounding. IEEE Transactions on Multimedia, 26, 4334-4347.

**Claims And Evidence:**

see above.

**Essential References Not Discussed:**

see above.

**Experimental Designs Or Analyses:**

see above.

**Methods And Evaluation Criteria:**

see above.

**Other Comments Or Suggestions:**

see above.

**Other Strengths And Weaknesses:**

see above.

**Questions For Authors:**

see above.

**Relation To Broader Scientific Literature:**

see above.

**Theoretical Claims:**

see above.

---

> ### Author Rebuttal · Authors · 2025-03-31
>
> ### **Q1 Writing**
> We will carefully revise the writing based on your suggestions. “CPCF” is an abbreviation formed from the initials of "Cross-Prompt Contrastive Framework". We will clarify this in the revised paper.
> ### **Q2 Figures**
> We will incorporate results of the image editing experiment into Figure 1. Figure 1 aims to highlight a common limitation of prior methods: generating wrong responses tailored to misleading regions that are adjacent or similar to the target object, which is the motivation for this work. Figure 8 in supp provides cases comparing our CPCF with the baseline Ferret, where CPCF produces correct responses while Ferret fails, showing the advantages of our method.
> ### **Q3 Related works about grounding and referring MLLMs**
> Thank you for pointing out these works! Different from them, our CPCF designs the first automatic contrastive decoding technique for referring MLLMs and with many task-specific designs tailored to this setting (see the answer for Q9 for details). We will cite and include a discussion of these papers in our revised paper.
> ### **Q4 Comparison with other self-training methods**
> Thank you for recommending these two works, **which are very excellent! We promise that we will cite them (CLIP-VG [4] and VG-Annotator [5]), and include corresponding discussions in our revised paper**.  Our method differs significantly from [4,5] in 2 key aspects:
> * **Different data Generation Methods**: [4,5] rely on templates, scene graphs, or small NLP models for generation, which may lack diversity in the generated data. In contrast, our CPCF leverages the more powerful MLLMs with **a novelly designed RAG framework**, where a similar labeled example is applied to guide the generation process for each unlabeled data. The stronger generative capabilities of MLLMs enhance data diversity, while RAG improves stability and mitigates errors.
> * **Different Training Strategies on Generated Data**: [4, 5] use conventional CE loss, L1 loss, or mIoU loss, to train the generated data. In contrast, our method introduces a carefully designed Direct Preference Optimization (DPO) framework, and the preference pairs in DPO are task-tailored designed—specifically, commonly generated vs. CoT-generated answers. **This is the first application of DPO to referring MLLMs (and with task-specific designs, not only simple usage)**, and it significantly outperforms CE loss (Table 6).
>
> With these innovations, we believe our self-training method is novel. Thank you!
> ### **Q5 Computational overhead**
> Please see our answer to Reviewer Xoy3 Q1. Thanks!
> ### **Q6 & Q7 Discussion with other CD methods (like CRG)**
> Please see our answer to Reviewer hUfo Q1. Thanks!
> ### **Q8 Effect of distillation on accuracy**
> Yes, distillation slightly affects accuracy, as shown in Table 4, where box accuracy and mask accuracy decrease by 0.71 and 0.60, respectively, after applying distillation. However, given that the distillation can largely reduce average inference time by 74%, we believe this minor loss in accuracy is a worthwhile trade-off and completely acceptable. More importantly, the distilled model still significantly outperforms the previous SOTA by +6.22 in box accuracy and +5.36 in mask accuracy, demonstrating that it remains highly effective despite the compression.
> ### **Q9 Innovations**
> Our framework contains multiple components, including contrastive decoding, self-training, and distillation. However, they are NOT simple combinations of existing modules, but each with the following key novelties:
> * (1) **Contrastive Decoding (CD)**: Prior CD methods typically rely on manually constructed contrastive objects (e.g., VCD uses perturbed images generated from random noise), which may introduce uncertainty and are difficult to be optimal (e.g., it is difficult to identify the most appropriate random noise for VCD). In contrast, **we are the first to propose an automated and learnable approach for identifying contrastive objects within the CD framework**, where a prompt extraction network is trained to find the optimal contrastive target **automatically**. This makes our method fundamentally different from previous CD methods and with better performance (Table 3).
> * (2) **Self-Training**: Kindly see our answer to Q4.
> * (3) **Distillation**: **We are the first to apply distillation to CD**, effectively addressing the issue of high computational cost in CD. More importantly, our framework does NOT simply adopt existing distillation techniques; instead, **we propose a novel distillation loss (Eq. 5, $\mathcal{L}_{inp}$) specifically tailored to the characteristics of CD and referring MLLMs**, which is highly effective (Table 8).
>
> In summary, **we have introduced task-specific, novel designs for each key component of our framework. Therefore, we believe our method is novel and can provide new insights**. We will clarify these more clearly in the revised paper and cite your recommended papers. Thank you!

---

> > ### Comment · Reviewer_avef · 2025-04-05
> >
> > Thanks to the authors for their efforts in the rebuttal. After reading the authors' response and considering my 9th comments, I decided to keep my review rating.

---

> > > ### Author Response · Authors · 2025-04-05
> > >
> > > Thank you again for your comments. However, we would like to reiterate that our method is **NOT** a simple combination of existing techniques, but instead incorporates several **novel designs specifically tailored to this task**, as we have detailed in our rebuttal. Specifically, our contrastive decoding method introduces a **newly proposed automatic contrastive target selection mechanism**; our self-training strategy adopts a **data generation and training method that is fundamentally different** from those in your mentioned [4, 5]; and our distillation framework includes a **new, task-specific loss function (Eq.5)**. More specifically, our method introduces significant improvements over previous contrastive decoding (CD) approaches in the following three key aspects: **working mechanism, training strategy, and efficiency enhancement**.
> > >
> > > * (1) **Regarding working mechanism: we propose the first automatic contrastive target selection method (sec 3.2) for CD framework**: Prior CD methods typically rely on manually constructed contrastive objects (e.g., VCD uses perturbed images generated from random noise), which may introduce uncertainty and are difficult to be optimal (e.g., it is difficult to identify the most appropriate random noise for VCD). In contrast, **we are the first to propose an automated and learnable approach for identifying contrastive objects within the CD framework**, where a prompt extraction network is trained to find the optimal contrastive target **automatically**. This makes our method fundamentally different from previous CD methods and with better performance (Table 3).
> > > * (2) **Regarding training strategy: we propose the first self-training method (Section 3.3) specifically designed to enhance the effectiveness of CD.** Moreover, this training method requires only unlabeled data, without the need for any manually annotated question–answer pairs, making it easy to collect in real-world applications. More importantly, our self-training strategy is fundamentally different from—and superior to—the approaches proposed in [4, 5] mentioned by Reviewer avef, both in terms of data generation and training methods. This is detailed in our response to Reviewer avef’s Q4.
> > > * (3) **Regarding efficiency enhancement: we propose the first distillation method specifically designed for contrastive decoding (CD), greatly reducing the inference time of previous CD methods by 73%**. More importantly, we do not simply adopt existing distillation techniques; instead, we design a novel distillation loss function (Eq. 5) specifically tailored to the unique characteristics of CD and referring MLLMs. This design proves to be highly effective, as evidenced by the results in Table 8.
> > >
> > > Incorporating these novel designs, **our proposed method is an entirely new CD framework tailored to referring MLLMs and with better performance (Table 3) and higher efficiency**. Thus, we believe that our method is novel.
> > >
> > > Additionally, we are pleased to note that **the novelty and contributions of our work have been recognized by many other reviewers**, such as **Reviewer Xoy3** (*"CPCF is a novel framework that leverages contrastive prompts to enhance referring MLLMs. The proposed framework is well-developed and incorporates several improvements that enhance both accuracy and efficiency."*), **Reviewer Vjf8** (*"The innovation of this article is quite clever."*), **Reviewer Za66** ("*The novelty lies in task-specific contrastive decoding, automated prompt selection, and efficiency-aware distillation. These are meaningful contributions to the literature.*"), and **Reviewer hUfo** (*"Innovative idea of leveraging misleading prompts in a structured contrastive setup"*).
> > >
> > > Thank you once again for the time and effort you devoted to reviewing our paper.

---

### Official Review · Reviewer_hUfo · 2025-03-24

**Overall Recommendation:** 2

**Summary:**

This paper introduces CPCF, a novel framework designed to enhance referring capabilities in MLLMs. The method leverages a cross-prompt contrastive strategy, in which responses generated from visual prompts are contrasted with those from misleading regions. The framework further incorporates a prompt extraction network to identify contrastive prompts, a self-training approach to synthesize training data from unlabeled images, and a distillation technique to reduce computational overhead.

**Claims And Evidence:**

The central claim is that CPCF improves the robustness and accuracy of referring MLLMs by mitigating the influence of misleading visual regions. While the empirical gains are consistent across benchmarks, some of the claims particularly regarding the “significant differences” from prior contrastive decoding methods like CRG, are overstated. The evidence for CPCF’s advantage over CRG largely hinges on controlled benchmarks and implementation-level choices, without a rigorous apples-to-apples comparison.

**Essential References Not Discussed:**

The paper does not cite work on robust grounding or saliency-based region selection, which could offer alternative approaches to handling misleading regions.

**Experimental Designs Or Analyses:**

While the experimental section is thorough in scope, some issues limit its interpretability:

- The baseline comparisons (e.g., with Ferret) are fair, but CRG, the most relevant contrastive approach, is not re-implemented with CPCF’s enhancements (e.g., distillation), limiting the claim that CPCF is “significantly different and better”.

- Appendix tables suggest that CPCF’s improvements often arise from better training data rather than the contrastive mechanism per se (see Table 7).

- The computational efficiency claims rely on distillation, but this step is itself complex and introduces extra modules (AGN, adapters). No wall-clock or energy consumption comparisons are provided.

**Methods And Evaluation Criteria:**

The methodology is generally well-motivated but not without caveats:

- The prompt extraction network is a reasonable architectural choice, but its complexity (e.g., priors like semantic similarity and distance maps, clustering, noise injection) introduces many design decisions with limited ablation or theoretical justification.

- The self-training method relies on synthetic question-answer generation, which raises concerns about data quality. Although the authors attempt to mitigate this via RAG-based retrieval and CoT prompting, the robustness of these synthetic annotations is unclear.

- The use of DPO loss with CoT vs. direct generation is under-motivated; it's unclear if simpler reward-based tuning methods would suffice.

- The evaluation tasks are standard, but most of the experimental results show improvements in the range of ~1–2%, which is modest given the complexity of the proposed pipeline.

**Other Comments Or Suggestions:**

No.

**Other Strengths And Weaknesses:**

Strengths:

- Innovative idea of leveraging misleading prompts in a structured contrastive setup.

- Distillation step is well-motivated to reduce inference cost.

- Strong empirical results on standard benchmarks.

Weaknesses:

- Significant complexity in the pipeline with many interdependent components.

- Marginal gains relative to added engineering overhead.

- Limited generalization claims beyond the specific tasks used in training/evaluation.

**Questions For Authors:**

- Can you isolate the impact of contrastive decoding without prompt extraction, self-training, or distillation?

- How robust are the gains when tested on out-of-distribution prompts or images?

- How would CPCF behave with overlapping or ambiguous prompts (e.g., in crowded scenes)?

**Relation To Broader Scientific Literature:**

The paper is aware of recent work in referring MLLMs and contrastive decoding. However, the differences with works like CRG and ICD are primarily empirical and implementation-level, not conceptual. The line between CPCF and these prior methods is thinner than the authors suggest.

**Theoretical Claims:**

No.

---

> ### Author Rebuttal · Authors · 2025-03-31
>
> **Q1 Discussion with other contrastive decoding (CD) methods (like CRG)**
>
> * Although both methods use CD, the contrastive targets are different. CRG constructs contrastive targets by removing the target region directly from the image, which may severely disrupt the image’s integrity—especially when the target region is large—potentially leading the model to capture noisy information during contrast. Differently, our method does not alter the image content, mitigating this issue and resulting in better accuracy.
> * CRG and most other CD methods significantly increase computational cost, as each contrastive object must be processed separately by the MLLM. Our CPCF is the first CD method to introduce a distillation technique to address this issue, reducing inference costs by 73%.
> * Even when CRG is equipped with our enhancements (self-training, distillation), its resulting accuracy of 76.74 on ROC-Mask still falls short of CPCF (79.55), further highlighting our advantages.
>
> **Q2 Ablation for components in prompt extraction network**
>
> These ablation studies are included in main paper Table 5. Thank you!
>
> **Q3 Robustness on synthetic data quality**
>
> Please see our answer to Reviewer nWKK Q2&Q3. Thanks!
>
> **Q4 Simpler reward-based tuning**
>
> Nice insight! It is indeed possible to use other reward-based tuning methods, but they still require preference pairs to compute rewards in DPO or to train a reward model in RLHF. Other methods typically rely on additional models (e.g., GPT) or human annotations to obtain these preference pairs, which can be costly and labor-intensive. This serves as our motivation for using the model itself to generate preference pairs for DPO in this work, which is more cost- and labor-efficient.
>
> **Q5 Improvement**
>
> Kindly note that **across all 12 metrics in the 4 benchmarks presented in Table 1 and Table 2, our method outperforms the second-best one by more than 2% in 9 cases—accounting for 75% of the total.** This shows the significant improvements of our method. Also note that our method incurs only very minimal additional computational cost during inference (just +4.3% than the baseline Ferret). The significant performance improvement achieved with such a small increase in inference cost further shows the advantage of our approach.
>
> **Q6 Improvement from data or method?**
>
> **Even without using the better extra dataset, our method can still achieve SOTA**. Using the original dataset $\mathcal{D}$ without the extra dataset $\tilde{\mathcal{D}}$ in our method achieves an accuracy of 74.55 on ROC-Box, which still significantly outperforms previous methods such as Ferret (71.71), Osprey (72.15), and Shikra (64.60). This indicates that **the better extra dataset only serves as a strategy for further enhancement, but is not the sole reason for our method’s strong performance**. The proposed network architecture and training strategy also play a crucial role in achieving high effectiveness.
>
> **Q7 Wall-clock of distillation**
>
> As indicated in Sec 4.1, distillation requires 25 hours of training for the 7B model on 8 A100 GPUs. Considering that this method reduces inference-time computation by 74% with only a very minimal drop in accuracy (see Table 4), we believe the additional training cost is worthy.
>
> **Q8 Pipeline complexity**
>
> Please see our answer to Reviewer WGMH's Q1. Thanks!
>
> **Q9 Contrastive decoding (CD) w/o other components**
>
> Good advice! We evaluate a method where all other components are removed, and contrastive prompts are manually sampled from regions adjacent to or similar to the target object for CD. This method achieves accuracies of 74.33 on ROC-Box and 75.08 on ROC-Mask, outperforming the baseline Ferret by 2.62 and 2.69, respectively, but still falling short of our full method by 4.04 and 4.47. These results demonstrate that both CD and other designed components are useful.
>
> **Q10 OOD prompts and images**
>
> * OOD prompts: We remove the data with scribble prompts and train the model using only other prompts. When evaluated with scribble prompts, this model achieves an accuracy of 73.09—lower than our fully trained model (77.97), but still higher than the fully trained Ferret (71.58).
>
> * OOD Images: Images for text recognition are NOT included during training, so the Referring Text Classification (RTC) task in Table 1 is an OOD setting. In this scenario, our method outperforms all previous methods, showing its strong generalization ability.
>
> **Q11 Overlapping or ambiguous prompts**
>
> Good advice! We select 422 crowded scene images with overlapping or ambiguous prompts. Our CPCF achieves an accuracy of 66.98, significantly outperforming Ferret (59.79), showing the high robustness of our method. These results will be included in the revised paper. Thanks!
>
> In summary, **CPCF significantly improves accuracy (more than 2% in 75% of all metrics), while requiring very minimal additional inference cost (+4.3% than Ferret)**.  We'll illustrate more clearly in paper. Thank you!

---

### Decision · Program_Chairs · 2025-05-01

**Decision:**

Accept (poster)

**Comment:**

This paper received mixed reviews, including two accepts, three weak accepts, and two weak rejects. Reviewers generally found the problem of improving referring MLLMs important and appreciated the paper’s well-structured empirical results, especially the performance gains achieved with modest inference overhead. The proposed CPCF framework includes automatic contrastive prompt extraction, self-training, and distillation techniques.

However, several reviewers raised critical concerns: (1) the method is a composition of existing components with limited novelty; (2) comparisons with key baselines such as CRG are incomplete or implementation-dependent; (3) claims of substantial improvements were challenged and, though rebutted with quantitative clarification, some reviewers remained unconvinced; (4) the overall system is complex, and the practical gains over previous contrastive decoding methods are incremental.

The rebuttal was detailed and addressed some concerns, but given the concerns about novelty and complexity, the AC recommends weak accept. The authors are encouraged to further highlight the novelty and consider simplifications for broader applicability.